# Multiple roles of H2A.Z in regulating promoter chromatin architecture in human cells

Lauren Cole[1,4], Sebastian Kurscheid [2,4], Maxim Nekrasov [2,4], Renae Domaschenz[2], Daniel L. Vera[1,3], Jonathan H. Dennis[1,5✉] & David J. Tremethick [2,5✉]

Chromatin accessibility of a promoter is fundamental in regulating transcriptional activity. The histone variant H2A.Z has been shown to contribute to this regulation, but its role has remained poorly understood. Here, we prepare high-depth maps of the position and accessibility of H2A.Z-containing nucleosomes for all human Pol II promoters in epithelial, mesenchymal and isogenic cancer cell lines. We find that, in contrast to the prevailing model, many different types of active and inactive promoter structures are observed that differ in their nucleosome organization and sensitivity to MNase digestion. Key aspects of an active chromatin structure include positioned H2A.Z MNase resistant nucleosomes upstream or downstream of the TSS, and a MNase sensitive nucleosome at the TSS. Furthermore, the loss of H2A.Z leads to a dramatic increase in the accessibility of transcription factor binding sites. Collectively, these results suggest that H2A.Z has multiple and distinct roles in regulating gene expression dependent upon its location in a promoter.

[1] College of Arts and Sciences, Department of Biological Sciences, Florida State University, Tallahassee, FL, USA. [2] The John Curtin School of Medical Research, The Australian National University, Canberra, Australia. [3] Present address: Department of Genetics, Blavatnik Institute, Paul F. Glenn Center for Biology of Aging Research, Harvard Medical School, Boston, MA, USA. [4] These authors contributed equally: Lauren Cole, Sebastian Kurscheid, Maxim Nekrasov. [6] These authors jointly supervised this work: Jonathan H. Dennis, David J. Tremethick. ✉email: dennis@bio.fsu.edu; David.Tremethick@anu.edu.au

The structure and dynamics of promoter chromatin is a principal driver of regulated gene expression[1–4]. The fundamental subunit of chromatin is the nucleosome core, ~150 base pairs of DNA wrapped 1.65 times around a histone octamer[5]. The prevailing functional view of a nucleosome (the nucleosome core plus linker DNA) is that it acts as a barrier preventing DNA access[6–8]. Therefore, the current model holds that the specific positioning of nucleosomes within a promoter plays an important role in controlling the accessibility of key regulatory DNA elements and the transcription start site (TSS) to transcription factors and RNA Pol II, respectively[1,2,9,10].

Genome-wide mapping of nucleosomes for several unicellular and multicellular organisms has provided an unprecedented amount of information concerning how nucleosomes are organised along the DNA of promoters, and how this organisation changes prior to and following transcriptional activation[10–20]. Such promoter mapping studies have revealed an apparent common nucleosome arrangement with a nucleosome free or depleted region (NFR/NDR) at an active transcription start site (TSS) followed by a high occupancy and strongly positioned '+1' nucleosome immediately downstream of the TSS. This strongly positioned +1 nucleosome then establishes a uniform or phased arrangement of nucleosomes downstream into the gene body. Similarly, in most cases, these nucleosome mapping studies revealed a strongly positioned '−1' nucleosome located immediately upstream of an active TSS. A repressed gene, on the other hand, lacks this phasing of nucleosomes around the TSS and, as expected, the TSS is also covered by a nucleosome[17,19]. Therefore, transcriptional activation has been proposed to involve nucleosome eviction from the TSS followed by the phasing of downstream and upstream nucleosomes[10].

It has been proposed that strong intrinsic nucleosome-repelling regions are encoded in the DNA sequence of gene promoters from simple unicellular organisms, such as budding yeast[21]. On the other hand, in humans and other complex multicellular organisms, promoters possess nucleosome attracting-regions[21]. This suggests that nucleosomes may be utilised in more complex ways to control the output of a promoter in complex organisms[22]. Indeed, new data are starting to emerge that challenge the current model; more recent information suggests that, rather than removing or repositioning a nucleosome to gain DNA access, the stability of a nucleosome may be modulated to enhance DNA accessibility[23].

Current approaches for mapping nucleosome occupancy often employ micrococcal nuclease (MNase) to produce mononucleosome-size DNA fragments (~150 bp) for subsequent sequence analysis. However, the majority of studies to date have only used a single digestion time point[1]. It is well established that the biochemical properties of nucleosomes vary significantly throughout the genome, which would affect their inherent stability and thus their sensitivity to MNase digestion[5,24]. In other words, a single level of MNase digestion may over-digest active unstable nucleosomes or under-digest stably repressed chromatin[1]. This is illustrated by the recent observation that the TSS may in fact not be nucleosome-free but contain a highly unstable nucleosome, which was not detected earlier because of its extreme sensitivity to MNase digestion and/or salt conditions employed[10,25,26]. However, this remains controversial[27]. By employing more than one level of MNase digestion, we and others previously showed that, despite nucleosomes displaying a similar level of occupancy in promoters, they exhibited differential sensitivity to MNase digestion[28–30]. Combined with a more recent study[23], the emerging view is that high nucleosome occupancy does not necessarily exclude high accessibility.

Along these lines, it has become clear that the nucleosome is not a static structure but exists as a highly dynamic family of interconverting partially assembled or disassembled structural states[5,31]. The existence of such sub-nucleosomal particles would also play an important role in regulating the accessibility of genomic DNA in cells[28,32]. However, again, most studies to date have only selected the canonical ~150 bp nucleosomal-sized DNA fragment for high-throughput sequence analysis and not included smaller sized DNA fragments[1]. DNA fragments <150 bp in size derived from MNase digestion may arise from sub-nucleosomal particles or from the protection of DNA by non-histone DNA binding proteins such as transcription factors[28,30]. Therefore, sequencing sub-nucleosomal DNA fragments may be as important as sequencing canonical nucleosome DNA fragments for the understanding of promoter structure and function.

A major player in shaping the chromatin organisation of a promoter is the evolutionarily conserved histone variant H2A.Z[10,33]. However, despite considerable effort to understand the functional relationship between H2A.Z and transcription, its function remains enigmatic because the data supports both positive and negative roles in transcription[10,33]. To our knowledge, no high-resolution map of H2A.Z at mammalian promoters has been produced, but composite H2A.Z ChIP-seq studies in humans and mice have revealed the formation of two well-positioned H2A.Z nucleosomes flanking each side of the TSS (the −1 and +1 nucleosome)[34,35]. It also has been reported that an unstable H2A.Z-containing nucleosome occupies the TSS of an active promoter[25,35]. Also controversial is whether the +1 H2A.Z-containing nucleosome facilitates transcriptional elongation or creates a strong barrier contributing to the pausing of RNA Pol II in cells[2,36–40]. In vitro biochemical studies have shown that mammalian H2A.Z stabilises the nucleosome core and inhibits transcription[41–43].

Previously, we demonstrated that H2A.Z is a major regulator of the epithelial-mesenchymal transition (EMT), which is a critical de-differentiation process required for early mammalian development and malignancy[37]. Moreover, we found that a +1 H2A.Z-containing nucleosome could repress the expression of some key epithelial genes while other H2A.Z promoter nucleosomes were correlated with gene activation. However, given the limitation of this study as well as other mammalian ChIP-Seq experiments as outlined above, an understanding of the role of H2A.Z in EMT and promoter function in general will require high-resolution H2A.Z-nucleosome position and MNase sensitivity maps. Additionally, analyses that includes mapping sub-nucleosomal particles will create a more complete picture of the role H2A.Z in regulating promoter architecture and transcription.

Here, we have employed an MNase-Transcription Start Site Sequence Capture method (mTSS-seq) established previously[13] to produce high-resolution H2A.Z-containing (and total) nucleosome maps of a 2KB region surrounding the TSS of the (~22,000) open reading frames of the human genome. To begin to understand how H2A.Z can facilitate or inhibit transcription, we have conducted our study in the context of gene expression changes associated with different epithelial cellular states, utilised a differential MNase digestion approach to obtain information on nucleosome stability, and analysed sub-nucleosomal-sized fragments to examine non-canonical nucleosomal and regulatory factor binding. These experiments have allowed us to redefine the nucleosomal architecture of active and inactive promoters, characterise nucleosome dynamics under different physiological contexts, and define the multiple roles of H2A.Z in these processes.

## Results

**MNase-transcription start site sequence capture.** To gain new insights into the link between gene expression and promoter

architecture, we employed our previously developed mTSS-seq approach[13] to map both total MNase protected nucleosome footprints and specifically immunoprecipitated H2A.Z-containing nucleosomes as well as sub-nucleosomal particles. In addition, we correlated different promoter organisations with gene expression by RNA-Seq (Supplementary Fig. 2 and Supplementary Data 1).

mTSS-seq combines in-solution targeted enrichment of 2 kb surrounding TSSs of 21,857 human protein-coding genes, as curated by NCBI RefSeq[44]. This cost-effective approach dramatically increases the sequencing depth and resolution by reducing the size of the genome for sequencing from 3.4 Gb to 40 Mb. In order to demonstrate the TSS specific enrichment achieved using our custom sequence capture array, we compared the mean coverage of captured H2A.Z ChIP-Seq (and mTSS-seq) data at the TSS with the coverage obtained from conventional, genome-wide H2A.Z ChIP-Seq (and mTSS-seq) using the same sequencing libraries (see Methods section). For this and subsequent experiments, we used MCF-10A cells (and three different physiological states of this cell line), a human mammary epithelial cell line that are normal and near diploid.

In order to demonstrate the performance of the mTSS-seq capture approach, libraries were subsampled to 5 and 10 million reads (Supplementary Fig. 1a, b, respectively). In addition, fully sequenced capture libraries were compared with genome-wide H2A.Z ChIP libraries (Supplementary Fig. 1c). Qualitatively, the coverage profiles of the capture libraries show a higher-resolution of H2A.Z promoter positioning and occupancy compared to H2A.Z genome-wide libraries (Supplementary Fig. 1a–c). Quantitatively, we obtained an ~18-fold enrichment of H2A.Z across all promoters using mTSS-seq compared to genome-wide H2A.Z ChIP-Seq (Supplementary Fig. 1d). The quantitative targeted enrichment of H2A.Z compared to a previously published nucleosome mapping data set is presented in Supplementary Fig. 1f.

**Promoters display multiple different types of nucleosomal arrangements that do not change in different cellular states.** To directly investigate the role H2A.Z plays in regulating promoter nucleosome architecture and stability, the expression of H2A.Z was inhibited (~8-fold) using the lentiviral vector shH2A.ZpLVTHM in MCF-10A cells[37] (Supplementary Fig. 3). To examine how the organisation of H2A.Z-containing promoters change in response to cellular phenotypic changes, MCF-10A cells were treated with TGF-β, a pleiotropic transforming growth factor that can promote EMT in numerous lineages of epithelial cells. Following induction with TGF-β, the epithelial state changes into a mesenchymal state. The knockdown of H2A.Z can mimic TGF-β in inducing a mesenchymal phenotype[37]. In addition, we employed the highly metastatic breast cancer cell line MCF-10CA1a, which was derived from the MCF-10A parental cell line[45]. For all cell lines, the major H2A.Z isoform expressed is H2A.Z.1 (H2AFV is expressed only at ~10% of the level of H2AFZ, and its expression is not affected by the shH2A.Z lentiviral vector, Supplementary Data 1).

The prevailing view of an active promoter is strongly positioned +1 and −1 nucleosomes, which establish the statistical positioning of nucleosomes downstream and upstream of the TSS[46]. To test this concept, first, we produced heat maps displaying total (Fig. 1a) and H2A.Z nucleosome (Fig. 1b) occupancy by k-means clustering whereby seven clusters were produced centred on the TSS. We based our empirical choice of seven clusters on the minimum number of clusters that generated robust promoter categories without producing redundant categories. Two levels of MNase digestion (light and heavy, see

Methods) were used and the nucleosome maps from both digestion time points were combined to produce these maps. The heatmaps provide information about common or different nucleosomal patterns for different types of promoter classes based on the nucleosome abundance at the −2, −1, +1 and +2 positions.

Unexpectedly, rather than the canonical strongly positioned +1/−1 nucleosomal arrangement, our clustering approach revealed the high occupancy of a single well-positioned nucleosome at different locations for total (Fig. 1a) and nucleosomes containing H2A.Z (Fig. 1b) in MCF-10A cells. For total nucleosomes, a single high-occupancy nucleosome is observed either at the +2 (cluster 1), +1 (cluster 2), −2 (cluster 5) or −1 (cluster 6) position (see average profiles left of each heat map, Fig. 1a). Cluster 3 displays high occupancy but poorly positioned (fuzzy) nucleosomes upstream of the TSS. Cluster 4 displays low nucleosome occupancy at the +1 and −1 position. The major feature of cluster 7 is the absence of any clear nucleosome organisation pattern that displays strongly positioned nucleosomes.

It is important to point out that while such single high-occupancy nucleosomes exist at different locations on different promoters, nucleosomes still occupy the other important positions in the same promoter class. For example, cluster 1 in MCF10A cells is characterised by a strong +2 positioned nucleosome, but it also has a nucleosome at the +1 position but its occupancy is lower (Fig. 1a).

Gene ontology analysis in MCF-10A cells reveals that some of these different nucleosome promoter organisations are associated with both overlapping (e.g. cellular metabolic processes) and differing biological processes. For example, cluster 1 is linked with mitotic cell cycle process, cluster 3 is associated with nucleic acid metabolism and clusters 5 and 6 are associated with ubiquitin-dependent processes and cellular localisation, respectively. Clusters 4 and 7 are linked with biological processes unrelated to epithelial function (detection of sensory perception and developmental processes, respectively; Supplementary Data 2).

To investigate whether the nucleosomal arrangements identified in MCF-10A cells are maintained or change when induced with TGF-β, treated with shH2A.Z, or in the malignant state, the K-means clusters defined in MCF-10A cells were applied to these different cellular states i.e. the same promoters are kept in the same cluster for all different physiological states (Fig. 1a). No major changes are observed indicating that the dominant feature of a promoter of having highly positioned nucleosome either at the −2, −1, +1 or +2 position is conserved between different cellular states that differ significantly in their phenotype.

Similar to the total nucleosome promoter clustering analysis (Fig. 1a), different promoter structures are observed that contain a singly positioned and high-occupancy H2A.Z-containing nucleosome (Fig. 1b). H2A.Z occupies the −1 (clusters 3 and 6, which, intriguingly, differs slightly in position), −2 (cluster 7) and +1 (cluster 4) positions in MCF10A cells. Other H2A.Z configurations exist where the overall H2A.Z occupancy is lower but some phasing is apparent (cluster 1) or where H2A.Z nucleosomes between the −1 and −2 positions are poorly positioned and blurry (cluster 5; Fig. 1b). The largest cluster of genes (cluster 2), lacks H2A.Z. These types of H2A.Z promoter organisations also appear to largely remain unchanged upon TGF-β treatment, and following malignant transformation (Fig. 1b).

Next, we wanted to know if H2A.Z-containing nucleosomes occupy the same or different positions compared to the total nucleosome positioning profiles. To address this question, we performed K-means clustering of H2A.Z nucleosomes in MCF10A cells dependent upon the total nucleosome clustering

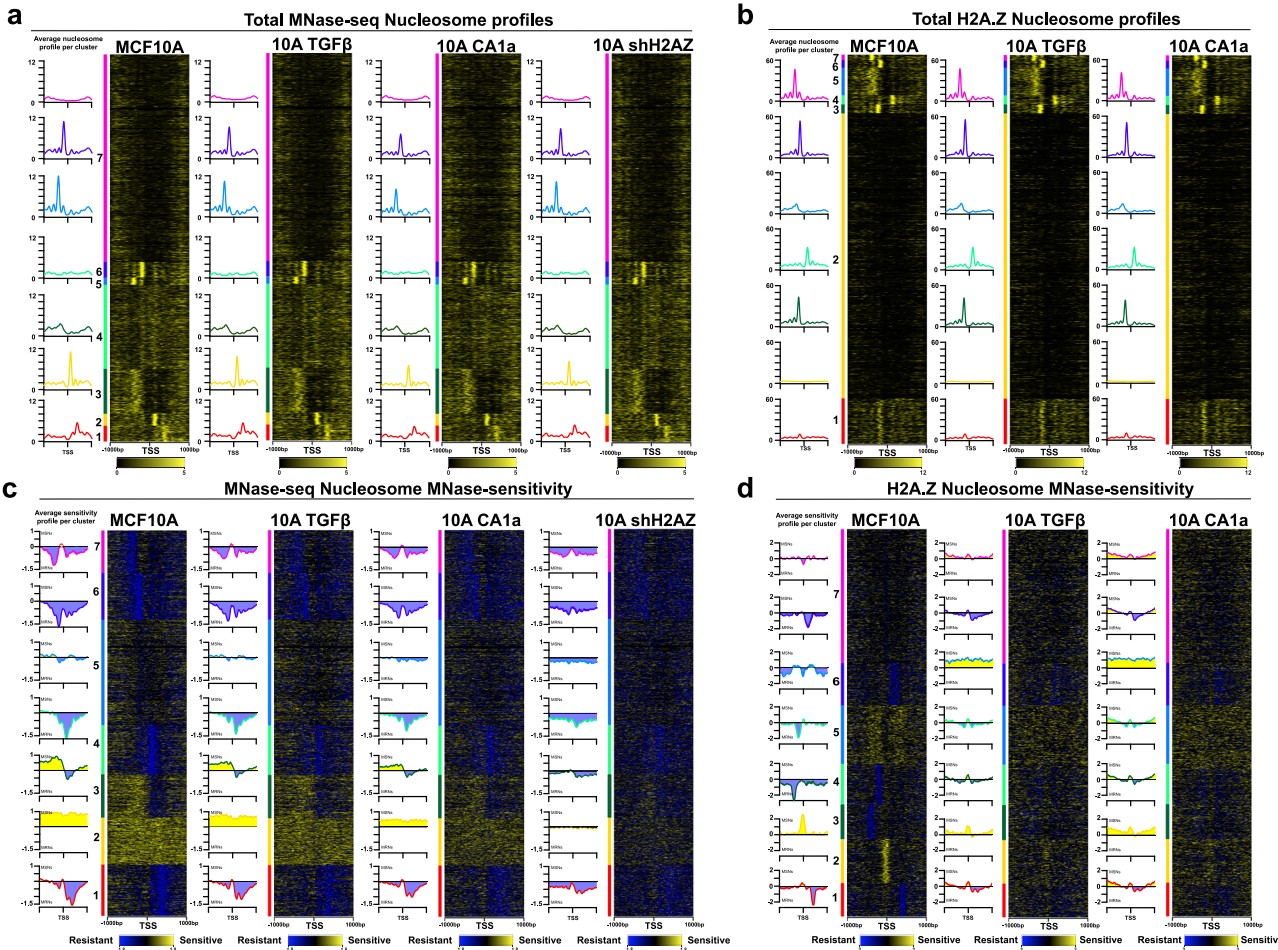

**Fig. 1 Comparison of the nucleosomal organisation and MNase sensitivity of all human promoters in epithelial, mesenchymal and cancer cell types. a** Heatmaps displaying total mTSS-seq nucleosome data for all human promoters centred on the TSS (±1 kb). Total nucleosome data for MCF-10A was sorted with k-means clustering ($k = 7$), followed by total nucleosome data for MCF-10A + TGF-β, MCF-10CA1a and shH2A.Z MCF-10A on the same sort order. Average profiles are shown to the left of the heatmaps for each cluster. The left most heatmap dictates the sort order for all other maps in the panel. **b** Heatmaps displaying total H2A.Z ChIP-seq data for all human promoters centred on the TSS (±1 kb). Total H2A.Z nucleosome data for MCF-10A was sorted with k-means clustering ($k = 7$), followed by total H2A.Z data for MCF-10A + TGF-β, and MCF-10CA1a on the same sort order. Average profiles are shown to the left of the heatmaps for each cluster. The left most heatmap dictates the sort order for all other maps in the panel. **c** The $\log_2$ratio of light/heavy MNase digest (MNase sensitivity) for total nucleosomes was determined for all human promoters centred on the TSS (±1 kb). The MNase sensitivity for the MCF-10A MTSS-seq data was sorted with k-means clustering ($k = 7$) followed by MNase sensitivity for MCF−10A + TGF-β, MCF-10CA1a and shH2A.Z MCF-10A on the same sort order. Average profiles are shown to the left of the heatmaps for each cluster. The left most heatmap dictates the sort order for all other maps in the panel. MNase sensitivity values for exemplar categories of highly positioned nucleosomes (cluster 6, −1 MRN and cluster 4, +1 MRN) are significantly different in MCF10A compared to each experimental condition (all p-values < $2.2 \times 10^{-16}$, using a two-sided Welch's t-test assuming unequal variances). **d** The $\log_2$ratio of light/heavy MNase digest (MNase sensitivity) for H2A.Z nucleosomes was determined for all human promoters centred on the TSS (±1 kb). The MNase sensitivity for immunoprecipitated H2A.Z-containing nucleosomes in MCF-10A was sorted with k-means clustering ($k = 7$) followed by the MNase sensitivity for H2A.Z MCF−10A + TGF-β, and H2A.Z MCF-10CA1a on the same sort order. Average profiles are shown to the left of the heatmaps for each cluster. The left most heatmap dictates the sort order for all other maps in the panel. MNase sensitivity values for exemplar categories of highly positioned nucleosomes (cluster 4, −1 MRN and cluster 2, MSN at TSS) are significantly different in MCF10A compared to each experimental condition (all p-values < $2.2 \times 10^{-16}$, using a two-sided Welch's t-test assuming unequal variances). The clustering window was set to ±500 bp surrounding the TSS for all analyses.

in Fig. 1a. The same H2A.Z nucleosome positions are observed as for total nucleosome positions indicating that canonical nucleosomes and H2A.Z-containing nucleosomes mostly occupy the same positions (Supplementary Fig. 4). However, subtle differences are observed e.g. H2A.Z cluster 1 is not observed in the total nucleosome profile nor is the slightly different H2A.Z nucleosomes positioned in the −1 position.

In conclusion, the high nucleosome depth afforded by the mTSS-seq approach reveals that many different promoter organisations exist, which display a single high-occupancy nucleosome located at different positions within the promoter.

Therefore, the classical view of a promoter, often based on low depth 'averaged' nucleosome promoter maps, that displays high-occupancy nucleosomes at the −2, −1, +1 and +2 positions may in fact be a composite of many different promoter organisations (see Discussion). Indeed, when we repeat the K-means clustering whereby two instead of seven clusters are produced, the classical view of a promoter with phased nucleosomes upstream and downstream of the nucleosome is recapitulated (Supplementary Fig. 5). The relationship between high nucleosome occupancy and gene expression is explored below.

**Promoters display multiple different types of MNase nucleosome sensitivity profiles that do change in different cellular states**. Next, we produced high-resolution heat maps to determine whether different MNase sensitivity profiles can define different promoter classes as determined by the log$_2$ ratio of light/heavy MNase digests. MNase sensitive and MNase-resistant nucleosomes will subsequently be referred to as MSNs and MRNs, respectively (these sorts were performed independently from the total nucleosome sorts in Fig. 1a, b).

Singly positioned MRNs can define different promoter classes (Fig. 1c). In MCF10A cells, a MRN is observed at the +2 position (cluster 1), +1 position (cluster 4), −1 position (cluster 6) and −2 (cluster 7) positions. On the other hand, non-MRN (cluster 5) and MSN promoter (clusters 2 and 3) classes exist but these nucleosomes are not positioned. Gene ontology analysis reveals again that different promoter classes are enriched for genes with different biological functions (Supplementary Data 2). Notably, cluster 2 contains gene ontology terms associated with brain function and behaviour e.g. sensory perception (and as expected, these promoters are transcriptionally silent, see below).

Next, we sorted the total nucleosome MCF10A data in the same sort order as the MCF10A MNase-sensitivity data. In clusters with dominant MSN or MRN positioned nucleosomes, positioned nucleosomes are observed across the entire promoter (as illustrated for the +2 MRN promoter cluster) (Supplementary Fig. 6). In other words, the dominant MSN or MRN positioned nucleosome does not necessarily correspond to the positioned nucleosome that displays the highest occupancy.

When the different MNase sensitivity promoter clusters for MCF-10A are directly compared with the clusters from the other cellular states, the largest change is observed when H2A.Z is depleted from cells. Specifically, there is a loss of MRNs in the different clusters (Fig. 1c, clusters 1, 4, 6 and 7; see adjacent average MNase sensitivity plots). This shows that H2A.Z is required for MNase resistance as demonstrated for the +1 nucleosome (Supplementary Fig. 7). These loci which lose MNase resistance at the +1 nucleosome in the H2A.Z knockdown contain 20.4% of the top significant differentially expressed genes between the MCF10A and shH2A.Z cells, as compared to 14.7% in a randomised sample of SeqCap genes (Supplementary Data 1). Curiously, the MSN clusters also lose sensitivity even though these promoter classes do not exist as a distinct H2A.Z MNase sensitive cluster (compare Fig. 1c with d). Taken together, major changes to the organisation of promoters occur in the absence of H2A.Z (see below).

Next, we repeated this analysis for the H2A.Z ChIP MNase sensitivity datasets in the MCF-10A, MCF-10A+TGF-β and MCF-10CA1a cells. Indeed, different promoter types are defined as having H2A.Z MRNs individually positioned at different locations. MRNs are located at the +2, −2, −1 and +1 positions (clusters 1, 3, 4 and 6, respectively; Fig. 1d). A distinct H2A.Z MSN positioned on the TSS is observed (cluster 2, Fig. 1d), which is not seen in the total MNase sensitivity profiles (Fig. 1c). H2A.Z MSNs are observed upstream and downstream of the TSS but these are also poorly positioned (and strongly repressed, see below; cluster 5). Strikingly, all promoter H2A.Z MRNs and MSNs classes change following the induction with TGF-β or transformation to the malignant state (Fig. 1d). This suggests that the promoter types in MCF-10A cells undergo dramatic changes in MNase sensitivity or resistance following changes in cell fate (Fig. 1c, d) but without major changes in nucleosome positioning (Fig. 1a, b).

Finally, we wanted to know whether these MSN and MRN promoter classes are specific for epithelial cells or are they more universal and exist in non-epithelial cell types. To examine this question, we repeated the mTSS-seq approach using the B-lymphoblastoid and lung fibroblast cell lines GM12878 and IMR90, respectively. We then applied the MCF-10A MNase sensitivity clustering analysis to both GM12878 and IMR90 data sets (Supplementary Fig. 8). This analysis revealed conservation of promoter classes between these desparate cell types and MCF-10A cells. Therefore, this shows that the different MSN and MRN promoter types observed in MCF-10A cells are not unique to this cell type.

In conclusion, different types of promoter classes are observed that contain singly positioned MRNs located at different positions within a promoter. On the other hand, MSNs are also present but these are poorly positioned with the exception of a H2A.Z MNase-sensitive nucleosome located at the TSS (see below). Moreover, promoter nucleosome MNase sensitivity but not nucleosome positioning changes following an alteration in cell fate. These results demonstrate that nucleosome positioning and MNase nucleosome accessibility are not coupled.

**High nucleosome occupancy, MNase resistance and a H2A.Z MSN at the TSS are features of active promoters**. Next, we investigated the relationship between total and H2A.Z nucleosome occupancy, and their respective MNase sensitivities, with gene expression by producing these four profiles for genes that are highly expressed (the top quartile of expressing genes, Fig. 2a). This analysis reveals that active promoters display: (1) high-occupancy nucleosomes (total and H2A.Z) that are positioned upstream and downstream of the TSS, (2) Micrococcal resistance across the promoter but in particular a MRN at the +1 position (total and H2A.Z, coloured blue in the average plots) and significantly, (3) a H2A.Z MSN at the TSS (Fig. 2a, coloured yellow in the average plot).

To link these features of active chromatin with the different promoter configurations observed for MCF-10A cells in Fig. 1, we independently sorted the four MCF10A datasets for the top quartile of expressed genes (Fig. 2a) into seven clusters (Supplementary Fig. 9). These clustering analyses recapitulates a subset clusters observed in Fig. 1 for total and H2A.Z nucleosomes and their respective MNase sensitivities. Specifically, the high occupancy of a singly positioned nucleosome and a positioned MNase-resistant nucleosome (upstream or downstream of the TSS), with a H2A.Z MNase-sensitive nucleosome occupying the TSS, are all reproduced.

To further confirm the relationship between nucleosome occupancy and positioning, we determined the level of expression for the different promoter classes in MCF-10A cells based on total and H2A.Z nucleosome occupancy (Fig. 1a, b). This analysis confirms a high level of expression is correlated with singly positioned nucleosomes (Supplementary Fig. 10). Conversely, poor nucleosome positioning (clusters 7 and 2 for total nucleosome and H2A.Z, respectively) is correlated with low levels of gene expression.

Given that major changes in the organisation of promoters into micrococcal resistant or sensitive chromatin structures occur following H2A.Z knockdown in MCF-10A cells, we tracked their chromatin architecture fate and gene expression changes by producing alluvial plots. K-means clustering was performed to produce MNase sensitivity heat maps of shH2A.Z MCF-10A cells independent of the original MCF-10A sort in order to identify the different types of promoter types that exist in these knockdown cells (Fig. 2b, c). First, we followed the fate of two inactive promoter classes in MCF-10A cells, the silent cluster 2 and the repressed cluster 5 (repressed cluster 5 displays nucleosomes that are neither MNase sensitive or resistant and hereafter will be referred to as MNase intermediate) that become active (Fig. 2b). Gene promoters from both clusters track to all types of active

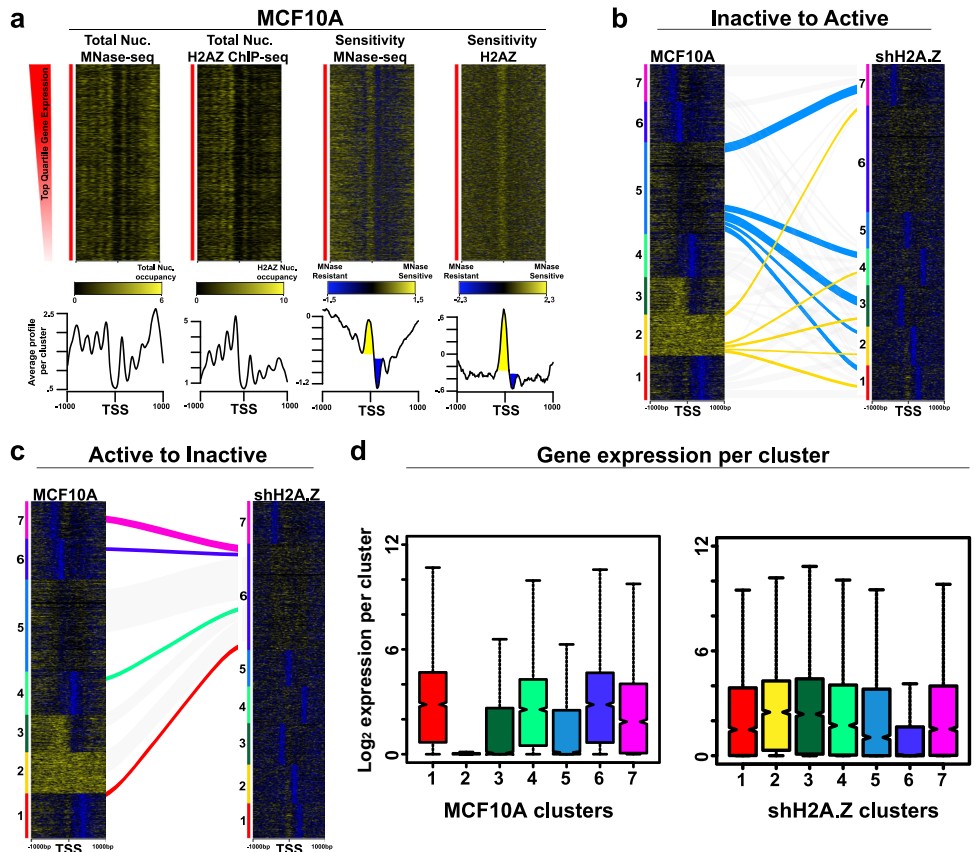

**Fig. 2 Promoters with high nucleosome occupancy and TSS MNase sensitivity are highly expressed. a** For the top quartile of expressing genes in MCF-10A cells, total mTSS-seq nucleosomes and MNase sensitivity, and immunoprecipitated H2A.Z nucleosomes and H2A.Z MNase sensitivity were sorted based on their expression. Average profiles are shown below. The yellow peak shows the MSN at the TSS. The blue inverted peak shows the MRN at the +1 position. **b** Alluvial diagram of how the silent cluster 2 and repressed cluster 5 change following their activation when H2A.Z is depleted from MCF-10A cells. **c** Alluvial diagram showing the fate of active promoter clusters following their inhibition of transcription when H2A.Z is depleted from MCF-10A cells. Line thickness is proportional to the number of promoters that change. **d** gene expression box plots for the different MCF-10A and shH2A.Z MNase-sensitive promoter classes. Box plots represent the data in quartiles with the median shown as a notch, with the middle 50% of the data contained in the box, and the minima and maxima shown with the upper and lower whiskers representing Q1 and Q4. RNA-seq was performed in three independent biological replicates.

promoter classes (clusters 1–5 and 7) but, a greater number of promoters from cluster 5 become active (Supplementary Data 3). Gene ontology analyses shows that genes that become activated are associated with EMT consistent with our previous study (Supplementary Data 4).

Next, we followed the fate of active promoter classes in MCF-10A cells (clusters 1, 4, 6 and 7; Fig. 2c) that become inactive (Supplementary Data 3). Such active promoters lose their micrococcal nuclease resistance and all track into cluster 6 (the MNase intermediate cluster) in shH2A.Z MCF-10A cells. These major alterations in promoter MNase accessibility, as shown by promoters from one type of MNase sensitivity arrangement moving to a different MNase sensitivity cluster, can account for the major chromatin organisational changes observed in Fig. 1c upon H2A.Z knockdown.

**Cell state-dependent changes in H2A.Z promoter MNase accessibility.** Given the induction with TGF-β or conversion to the malignant state alters the promoter organisation of all H2A.Z MRNs and MSNs classes (Fig. 1d), we tracked the fate of these promoter classes following these transformations. First, we followed the fate of promoters in a highly repressed promoter class (cluster 5, which has poorly positioned H2A.Z MSN centred around the −1 and +1 position) and the MNase intermediate-

repressed promoter class 7 in MCF-10A cells that become active (clusters 1, 2 and 6) following TGF-β treatment. As observed following the knockdown of H2A.Z expression, the majority of promoters that significantly increase in gene expression arise from the MNase intermediate-repressed promoter cluster and track to all the different active promoter types (Fig. 3a and Supplementary Data 3). Conversely, following the fate of active promoters in categories 1, 3, 4 and 6 in MCF-10A cells that significantly decrease in expression also primarily end up in the MNase intermediate-repressed cluster 7 (Fig. 3b and Supplementary Data 3).

When the fate of active or inactive promoter classes are similarly tracked upon malignant transformation (when MCF-10A cells are transformed into MCF-10CA1a cells), comparable results are obtained to that of TGF-β induction (Fig. 3c, d and Supplementary Data 3). In other words, it appears that it is the repressed MNase-intermediate promoter class that can be more easily remodelled to, or from, an active promoter configuration when the phenotype of a cell changes. Gene ontology analyses shows that genes whose expression is altered is associated with cancer progression (Supplementary Data 4).

**Active promoter classes are marked by active histone modifications.** Having defined a variety of different types of active

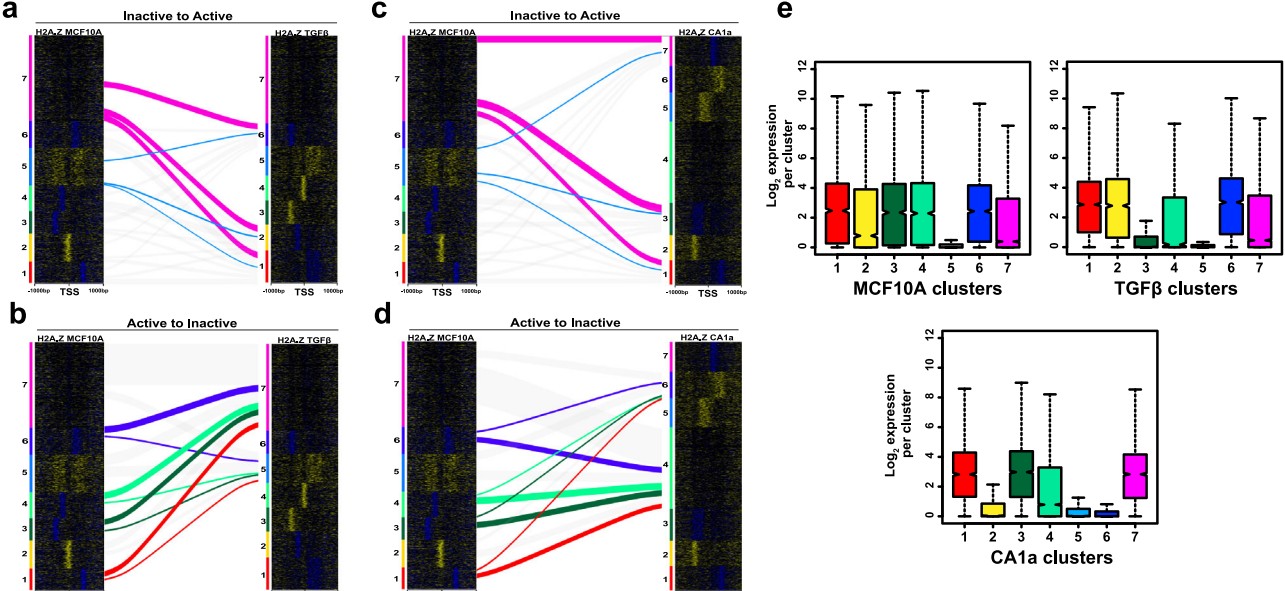

**Fig. 3 How different promoter H2A.Z MNase sensitive or resistant structures in MCF-10A cells change following TGF-β treatment and malignant transformation. a** Alluvial diagram of how H2A.Z repressed clusters 5 and 7 change following their activation when MCF-10A cells are induced with TGF-β. **b** Alluvial diagram showing the fate of active promoter clusters to repressed clusters 5 and 7 following their inhibition of transcription when MCF-10A cells are induced with TGF-β. **c** Alluvial diagram of how H2A.Z repressed clusters 5 and 7 change following their activation when MCF-10A cells become malignant. **d** Alluvial diagram showing the fate of active promoter clusters to repressed clusters 4 and 6 following their inhibition of transcription when MCF-10A cells become malignant. **e** gene expression box plots for the different MCF-10A, MCF-10A + TGFβ and MCF-10CA1a MNase-sensitive promoter classes. Box plots represent the data in quartiles with the median shown as a notch, with the middle 50% of the data contained in the box, and the minima and maxima shown with the upper and lower whiskers representing Q1 and Q4. RNA-seq was performed in three independent biological replicates.

promoter structures, we next investigated their histone modification status. To do this, we used publicly available ChIP-Seq data for MCF10A cells obtained from NCBI's Sequence Read Archive (H3K4me1, H3K4me3, H3K79me2, H3K27ac, H2BK20ub, H3K9me3, H3K9ac, H3K36me3, H3K23ac and H4K8ac; Supplementary Fig. 11). Confirming the new types of active promoter configurations identified (Figs. 1 and 3 and Supplementary Fig. 9), high occupancy of positioned total (Supplementary Fig. 11a) or H2A.Z-containing nucleosomes (Supplementary Fig. 11b) are strongly enriched with the active promoter histone modifications H3K4me3, H3K27ac, H3K9ac as well as displaying H3K4me1 and H4K8ac. Likewise, total (Supplementary Fig. 10c) and H2A.Z-containing active MRN promoter classes, plus promoters with a H2A.Z MNase-sensitive nucleosome at the TSS (Supplementary Fig. 10d), also display these active histone modifications. Conversley, these active modifications are depleted from the different types of strongly repressed promoter chromatin structures e.g. the total and H2A.Z-MSN clusters (clusters 2 and 3, and cluster 5, respectively; Supplementary Fig. 11). The moderately repressed clusters e.g. the total and H2A.Z-intermediate nucleosome sensitivity in clusters 5 and 7, respectively, display intermediate levels of active modifications because there are genes within these promoter classes that are low to moderately expressed (Figs. 2d and 3e).

**Loss of H2A.Z generates subnucleosomal-sized DNA fragments at the TSS.** To further explore the intriguing finding that a H2A.Z MSN occupies an active TSS in MCF-10A cells, we sorted the H2A.Z MNase sensitivity data at ±50 bp surrounding the TSS to resolve sensitivity patterns more precisely over the TSS (Fig. 4). This analysis revealed three distinct promoter configurations; TSSs that either have a H2A.Z-MSN (red cluster 1), a H2A.Z-MRN (green cluster 2) or no H2A.Z nucleosome at all (blue cluster 3; Fig. 4a). Notably, for these three different clusters, H2A.

Z-MSNs are associated with the highest level of gene expression, while the presence of a H2A.Z-MRN at the TSS is correlated with the lowest level of gene expression. The absence of an H2A.Z-containing nucleosome at the TSS displays an intermediate level of expression. The genes in each of the three clusters demonstrate significantly different levels of gene expression from each other (Fig. 4b). The high expressing MSN-TSS cluster displays strong nucleosome positioning (H2A.Z and total nucleosomes) downstream (and upstream) of the TSS whereas the repressive MRN-TSS cluster displays less strong ('fuzzier') nucleosome positioning with the average position of the +1 nucleosome being located closer to the TSS (Fig. 4a). Therefore, significant differences in the chromatin structure are observed between a H2A.Z MSN -TSS active state and a less active H2A.Z MRN -TSS state.

H2A.Z and total DNA fragment size distribution analysis for clusters 1 and 2 reveals the expected nucleosomal ~145 bp sized fragments at and surrounding the TSS (Fig. 4c). On the other hand, such sized fragments are not observed in cluster 3 which lacks a H2A.Z nucleosome at the TSS. Unexpectedly, upon H2A.Z knockdown, there is a dramatic increase in small sized DNA fragments (SFs) at the TSS in cluster 1 (and a modest increase at cluster 2). This suggests the H2A.Z-containing nucleosome, in particular the H2A.Z-MSN, is further destabilised upon the loss of H2A.Z, resulting in an increase in MNase accessibility. *TIMP1* and *S100A3* are examples of promoters that have H2A.Z nucleosomes at the TSS, leading to a dramatic increase in SFs following H2A.Z knockdown (Fig. 4d). These genes are involved in cancer progression[47,48] and their expression increases following H2A.Z knockdown (Supplementary Data 1).

**The presence of sub-nucleosomal sized particles at the TSS is correlated with higher levels of gene expression.** To further investigate the link between the generation of SFs at the TSS and gene expression, we aligned all SFs (<125 bp) to the TSS for

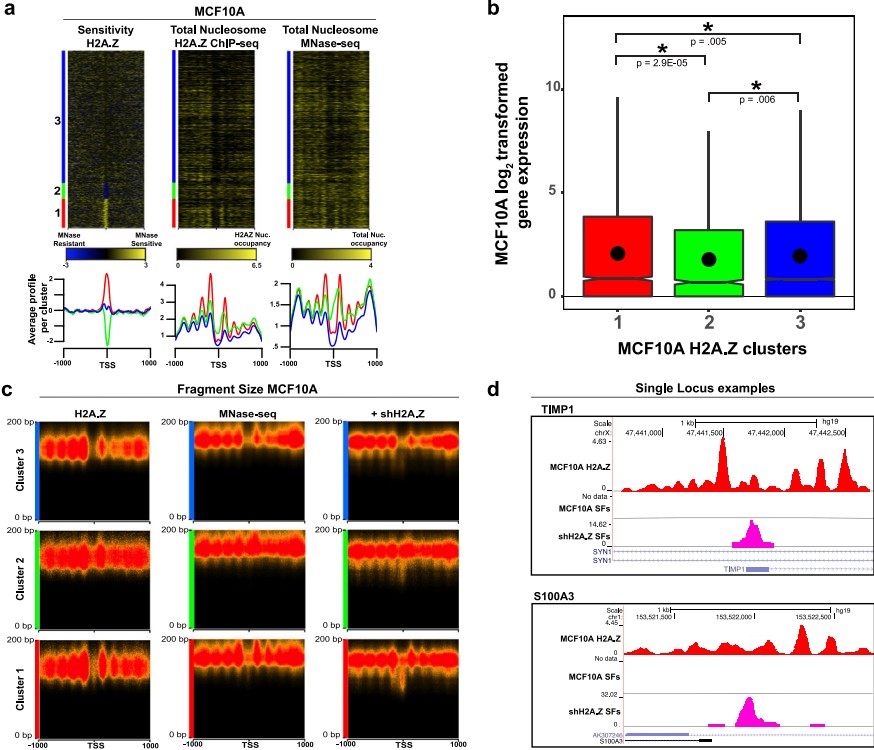

**Fig. 4 The MNase accessibility of the H2A.Z nucleosome at the TSS correlates with the level of expression. a** H2A.Z MNase sensitivity data was sorted with a clustering window ±50 bp surrounding the TSS by k-means clustering (*k* = 3) to produce a heat map that reveals H2AZ MNase resistant and sensitive nucleosomes at the TSS in MCF-10A cells. Adjacent heat maps display H2A.Z and total nucleosome profiles dependent on the H2A.Z MNase sensitivity sort. Below each heatmap are average nucleosomal occupancy profiles. **b** Gene expression box plots for each TSS H2A.Z MNase sensitivity cluster. *P*-values determined by two-tailed Welch's t-test assuming unequal variances. Box plots represent the data in quartiles with the median shown as a notch, with the middle 50% of the data contained in the box, and the minima and maxima shown with the upper and lower whiskers representing Q1 and Q4. RNA-seq was performed in three independent biological replicates. **c** Total MNase digestion fragment size distribution for each respective cluster in panel a, centred on the 1000 bp surrounding the TSS. DNA fragment size analysis was performed on immunoprecipitated H2A.Z nucleosomes, total MTSS-seq nucleosomes, and shH2A.Z nucleosomes for MCF10-A cells. Following H2A.Z knockdown, a distinct population of subnucleosomal-sized DNA fragments is observed in the red and green clusters 1 and 2, respectively. **d** Representative promoters, *TIMP1* and *S100A3* demonstrating the appearance of subnucleosomal-sized DNA fragments at the TSS following H2A.Z knockdown.

MCF-10A cells, MCF-10A cells induced with TGF-β, MCF-10CA1a cells, and shH2A.Z MCF-10A cells, respectively (Fig. 5a). Significantly, for each of these different cellular states, SFs can be detected at the TSS, and moreover, the top quartile of promoters containing these SFs (purple cluster 4) is correlated with the highest level of gene expression. Further, the knockdown of H2A. Z dramatically increases the proportion of SFs at the TSS (as observed in Fig. 4c) leading to even higher levels of gene expression. Therefore, the appearance of SFs at the TSS appears to be an unexpected feature of an active promoter.

Next, we examined whether these SFs originate from the same promoters in all the different cellular states, or are generated from different promoters in a cell type specific manner (Fig. 5b). Differential small-DNA-fragment heat maps (MCF-10A versus MCF-10A cells induced with TGF-β, MCF-10A versus MCF-10CA1a cells, and MCF-10A versus shH2A.Z MCF-10A cells) reveal that SFs originate from different promoters in a cell-type specific manner. In all cases, cluster 6 (pink) identifies promoters where there are more small DNA fragments in MCF-10A cells, while cluster 1 (red) identifies small DNA fragments that are specifically generated from TSSs following TGF-β induction, in MCF-10CA1a cells, and following H2A.Z knockdown, respectively (Fig. 5b).

In each different experimental condition, cluster 1 contains promoters that have an increase of SFs at the TSS. In TGF-β induced cells, this cluster contains 22% of the top Signficantly differentially expressed genes as compared to MCF10A. In

MCF-10CA1a cells, this cluster contains 16% of the top Signficantly differentially expressed genes as compared to MCF10A. Finally, in the H2A.Z knockdown cells, this cluster contains 21% of the top differentially expressed genes as compared to MCF10A. Cluster 6, which represents promoters that contain more SFs in the MCF10A cells, but less SFs than in each experimental condition, contains only 9–12% of the top significantly differentially expressed genes.

To further explore the correlation between changes gene in expression and the increase in SFs following H2A.Z knockdown, we identified common promoters among: (1) the top quartile of promoters with the most MNase-sensitive nucleosome at the TSS (5454 genes); (2) the top quartile of promoters that yield the most SFs at the TSS following H2A.Z knockdown (5,450 genes) and (3) the top 1% of genes whose expression increases following H2A.Z knockdown (1348 genes). This revealed that 110 promoters share all of these characteristics. A gene ontology analysis of these genes shows that they are indeed involved in epithelial related functions such as epidermal cell and keratinocyte differentiation, and regulation of morphogenesis of an epithelium (Fig. 5c).

**Loss of H2A.Z dramatically increases the population of subnucleosomal DNA fragments at transcription factor binding sites.** SFs derived from MNase digestion can arise from either the internal digestion of an unstable nucleosome core particle and/or by the protection of DNA by non-histone proteins such as

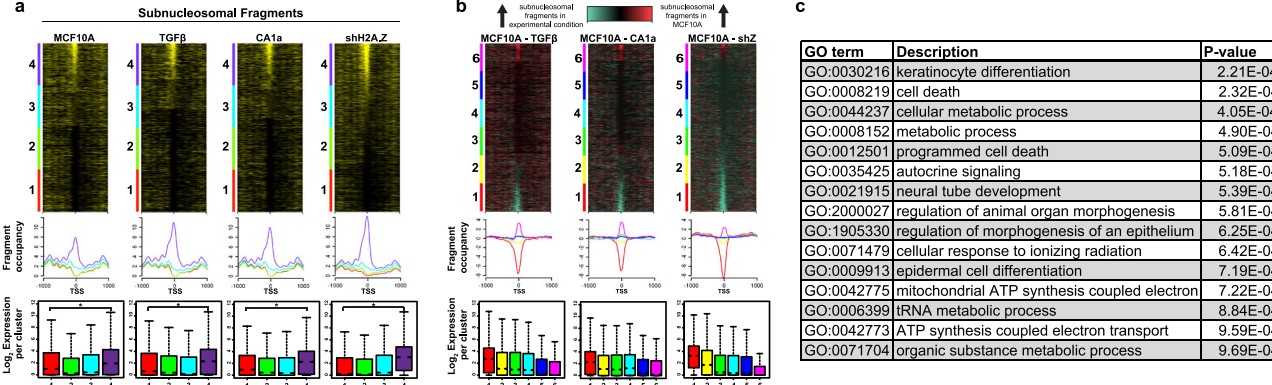

**Fig. 5 Subnucleosomal-sized DNA fragments at the TSS is correlated with high levels of gene expression. a** Subnucleosomal DNA fragments (<125 bp) were extracted and sorted for the maximum occupancy on 100 bp surrounding the TSS for MCF-10A, MCF-10A+TGF-β, MCF-10CA1a and shH2A.ZMCF-10A cells to produce heatmaps. The heatmaps were divided into quartiles from high to low occupancy. Below each heatmap are average subnucleosomal DNA fragment occupancy and gene expression box plots for each quartile. Box plots represent the data in quartiles with the median shown as a notch, with the middle 50% of the data contained in the box, and the minima and maxima shown with the upper and lower whiskers representing Q1 and Q4. RNA-seq was performed in three independent biological replicates. **b** Total subnucleosomal fragments from each experimental condition, MCF-10A+TGF-β, MCF-10CA1a cells and shH2A.ZMCF-10A, were each independently subtracted from total subnucleosomal fragments in MCF-10A and the resulting difference matrix was used to produce a differential heatmap. Red and cyan coloured SFs at the TSS indicate more abundant subnucleosomal DNA fragments in MCF-10A cells or more abundant subnucleosomal DNA fragments in the matched experimental condition, respectively. Below each heatmap are average subnucleosomal DNA fragment occupancy and gene expression box plots for each sextile. Box plots represent the data in quartiles with the median shown as a notch, with the middle 50% of the data contained in the box, and the minima and maxima shown with the upper and lower whiskers representing Q1 and Q4. RNA-seq was performed in three independent biological replicates. **c** Gene ontology analysis of 110 genes of interest using the GOrilla algorithm, *p*-values are determined by hypergeometric null model as described in Eden et al.[65].

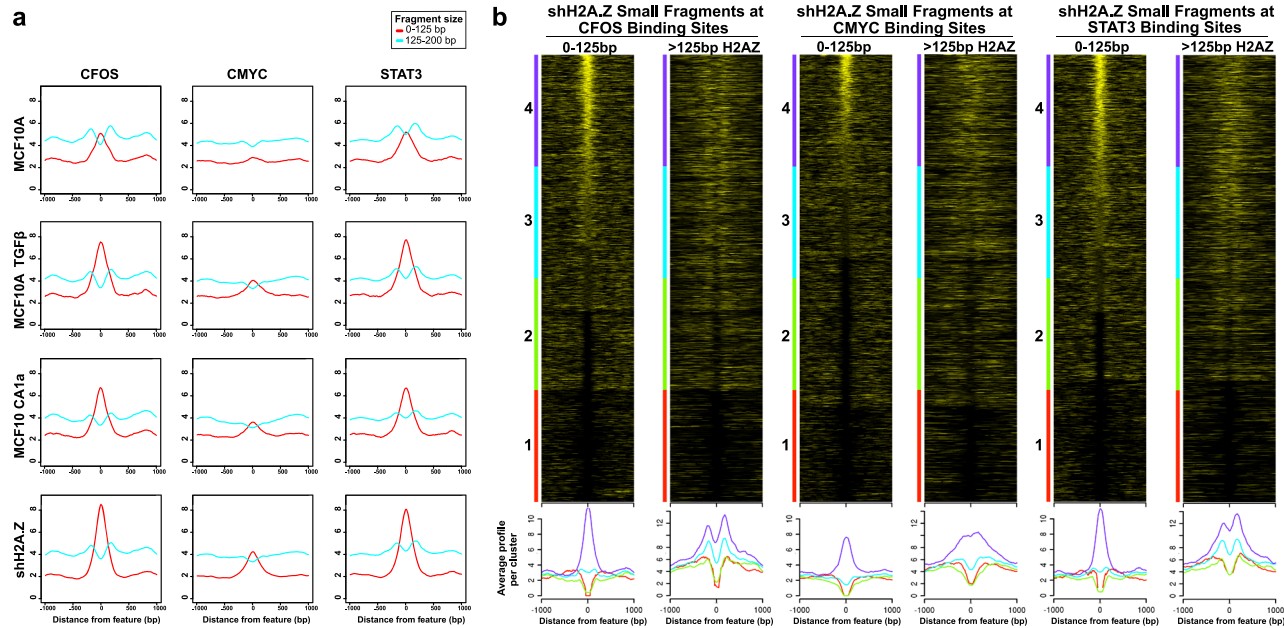

**Fig. 6 Mapping of small MNase DNA fragments may serve as a surrogate for transcription factor binding. a** Sub-nucleosomal small DNA fragments (<125 bp) and nucleosomal-sized DNA fragments (>125 bp) were extracted and aligned to C-Fos, C-Myc and STAT3 transcription factor binding sites for MCF-10A, MCF-10A+TGF-β, MCF-10CA1a and shH2A.ZMCF-10A cells. **b** Heatmaps, with average profiles below, were sorted on maximum occupancy of small DNA fragments (<125 bp) ±1 kb centred on the transcription factor binding site. Adjacent to each small DNA fragment heat map are H2A.Z nucleosome heatmaps sorted on the same order. H2A.Z nucleosomes flank C-FOS and STAT3 binding sites.

transcription factors. Our final aims were to investigate: (1) whether SFs occupy other important promoter DNA elements i.e. transcription factor binding sites (TFBS) and (2) if the abundance of these SFs also increase when H2A.Z is depleted.

To carry out these aims, we first utilised the mTSS-seq approach to map SFs at known transcription factor binding sites at high resolution. Specifically, we mapped <125 bp and >125 bp

DNA fragments aligned to the centre of previously identified c-Fos, c-Myc and STAT3 transcription factor binding sites[49] (Fig. 6a).

Clear SF peaks are observed at c-Fos and STAT3 binding sites, but the accumulation of these small DNA fragments at c-Myc binding sites is less obvious in MCF-10A cells (Fig. 6a). This supports the notion that mapping MNase-resistant SFs may serve

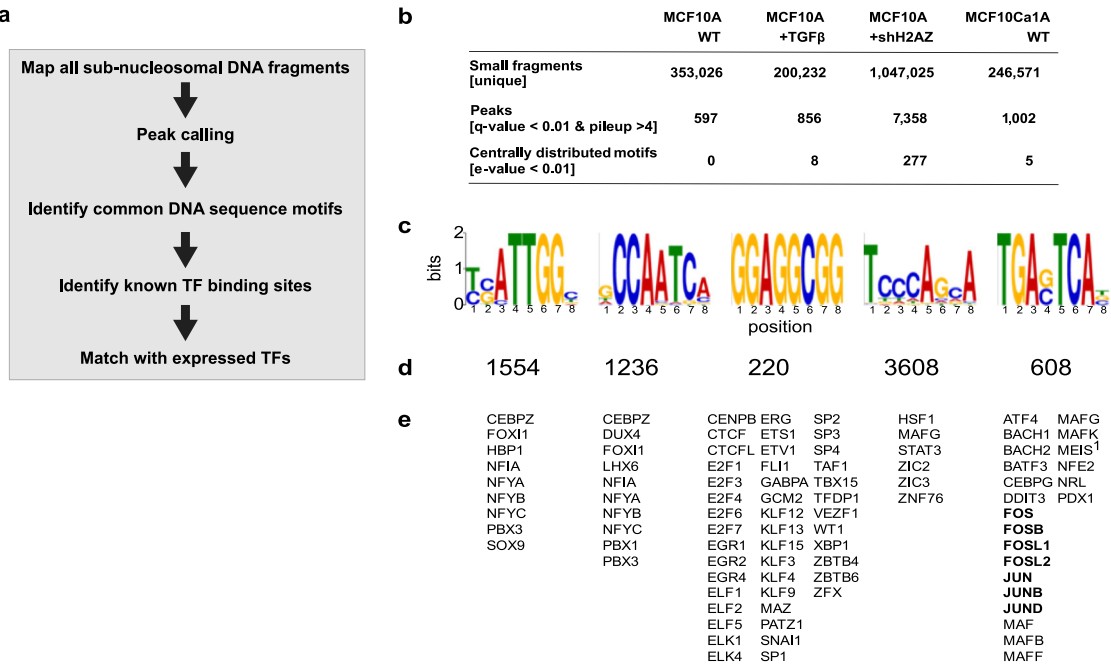

**Fig. 7 Loss of H2A.Z causes a dramatic increase in subnucleosomal-sized DNA fragments that map to transcription factor binding sites. a** Summary of the analysis. **b** Subnucleosomal-sized DNA fragments from all reads were extracted and aligned to all promoter regions and peak-calling was performed to detect enriched DNA sequence motifs within ±250 bp of the peak centres. **c** The top five DNA binding motifs that correspond to transcription factors expressed in the shH2A.Z MCF-10A cells. **d** The corresponding number of promoter sites at which each of these top five motifs were found in shH2A.Z MCF-10A cells. **e** The 93 transcription factors expressed, which recognise each of the top five DNA binding motifs in shH2A.Z MCF-10A cells. Highlighted in bold are important EMT transcription factors whose DNA recognition motif overlaps with subnucleosomal peaks that were previously occupied by H2A. Z nucleosomes.

as a surrogate for transcription factor binding. A pair of nucleosomes (>125 bp) flank either side of the c-Fos and STAT3 binding site, which, therefore, appear to create a boundary for these transcription factors (Fig. 6a, b). This is not observed for the c-Myc binding site. Following induction with TGF-β, in MCF-10CA1a cells or when the expression of H2A.Z is inhibited, the apparent binding of all three transcription factors increases, with the greatest increase occurring when H2A.Z is depleted from the cell (Fig. 6a). This is unrelated to the expression of these transcription factors; all three transcription factors are expressed at the highest level in MCF-10CA1a cells (Supplementary Fig. 2). Therefore, the loss of H2A.Z appears to allow the binding of these transcription factors. This can be explained if H2A.Z-containing nucleosomes that flank these TFBS are more refractory to the binding of transcription factors than canonical nucleosomes.

To address this, we sorted the MNase-produced SFs from high to low abundance centred at these transcription-factor binding sites, and aligned these sorts with H2A.Z nucleosome ChIP-Seq data (Fig. 6b). Indeed, the presence of SFs at c-Fos and STAT3 binding sites is correlated with the presence of a pair of H2A.Z nucleosomes that flank either side of these TFBS (Fig. 6b). On the other hand, H2A.Z nucleosomes are not well positioned at the c-Myc binding site (Fig. 6b). Notably, the formation of these nucleosomes that flank c-Fos and STAT3 binding sites cannot be dependent on H2A.Z, because they are still present when the expression of H2A.Z is inhibited (Fig. 6a). Rather, consistent with the above proposal, perhaps the loss of H2A.Z creates an even more accessible C-Fos and STAT3 binding site (see below).

Next, we investigated whether other TFBS could be identified de novo by mapping the location of SFs promoter-wide, and if these SFs also increased in the absence of H2A.Z (Fig. 7). To do this, we: (1) extracted all SFs from all reads and aligned them to

all promoter regions; (2) performed peak-calling to detect enriched sites within promoter regions, (3) identified common DNA sequence motifs (8–15 bp) within ±250 bp of the peak centres; (4) identified which of these common DNA sequence motifs are known human transcription factor binding sites, and finally; (5) determined which transcription factors are expressed, and matched these with the identified transcription-factor DNA binding motifs (Fig. 7a).

The number of SF peaks found in the promoters in MCF-10A induced with TGF-β (856) and MCF-10CA1a (1002) increased compared to MCF-10A cells (597). However, the most dramatic increase is found in promoters in shH2A.Z MCF-10A cells (7358; Fig. 7a), consistent with the proposal that the loss of H2A.Z increases transcription factor binding (Fig. 7a). Notably, 277 different but common DNA sequence motifs were identified in shH2A.Z cells (Fig. 7b). Of these 277 different DNA sequence motifs, 147 were known transcription-factor binding sites. The top 5 transcription factor binding sites (Fig. 7c) and the number of promoter sites at which each of the top 5 transcription DNA binding motifs were found in shH2A.Z MCF-10A cells are shown (Fig. 7d). Of the 147 known transcription factor binding sites, 93 motif-matched transcription factors are expressed in H2A.Z knockdown MCF-10A cells (Fig. 7e). Included in the top 5 transcription DNA binding motifs with matching transcription-factor expression is STAT3, the Fos and Jun family of transcription factors and the key EMT regulator SNAI1 (Fig. 7c, e).

Given that SFs arise from a H2A.Z nucleosome positioned on the TSS, we wondered whether the generation of these SFs at TFBS also arise from previously occupied H2A.Z-containing nucleosomes. We identified 5506 H2A.Z MSN promoter peaks and, significantly, 1859 SF peaks overlap with these H2A.Z peaks following H2A.Z knockdown (~34%, see Methods section). In

other words, a substantial proportion of SFs arise from sites previously occupied by H2A.Z nucleosomes. More striking, 100% of these SF peaks overlap with transcription-factor binding sites which have the matched transcription factor expressed (a total of 103 transcription factors, Supplementary Data 5). These transcription factors indeed include the Fos and Jun family and SNAI1 (Fig. 7e and Supplementary Data 5). Collectively, these observations strongly suggest that H2A.Z nucleosomes are located at TFBS to prevent promiscuous transcription factor access.

## Discussion

In this study we provide insights into the relationship between promoter nucleosome organisation and gene expression by generating, to our knowledge, the first high-depth H2A.Z and sub-nucleosomal maps, as well as determining their respective sensitivities to MNase digestion, for essentially all human RNA Polymerase II promoters (which will provide a valuable resource). The major H2A.Z isoform expressed in the different cell lines examined here is H2A.Z.1. Our data suggests that H2A.Z may have several roles in regulating the accessibility of promoters dependent on where it is located within a promoter e.g. upstream or downstream of the TSS, at the TSS or at transcription factor binding sites.

Specifically, we show that: (1) there are many different types of H2A.Z-containing active and inactive promoters structures that differ in their nucleosome organisation and sensitivity to MNase digestion; (2) a surprising feature of different types of active promoters is the presence of MRNs nucleosomes that are positioned either upstream or downstream of the TSS including a H2A.Z MRN at the +1 position. Other types of active promoters have a H2A.Z MSN occupying the TSS while less active and repressed promoters contain a H2A.Z MRN at this position. All the different types of active promoter nucleosomal configurations display the expected active histone marks thus confirming our classification of the different types of active and inactive promoter nucleosome arrangements; (3) the loss of H2A.Z results in a dramatic increase in the formation of SFs that map to important functional DNA elements (TFBS and the TSS). This suggests that H2A.Z has an important role in preventing promiscuous DNA access and (4) while promoter nucleosome positioning does not change between different cellular states, nucleosome MNase accessibility does, perhaps suggesting the latter feature is more important in regulating promoter activity.

Concerning the nucleosome arrangement of an active promoter, one important observation is that, rather than a single consensus active promoter organisation with a high occupancy −1 and + 1 nucleosome flanked by phased nucleosomes (Supplementary Fig. 5), there exist multiple different types of nucleosomal organisations characterised, for the most part, by a single high-occupancy nucleosome positioned at the −2, −1, +1 or +2 position. This arrangement is observed both for H2A.Z and total nucleosomes. An important question to address in the future is whether this characteristic of different active promoter configurations is dependent upon the underlying promoter nucleosome DNA sequence. However, we do know that when the DNA sequence of all positioned nucleosomes is analysed, the characteristic 10 bp periodicity of the AT dinucleotide is observed, but H2A.Z appears to be less dependent on this AT rotational phasing signal compared to bulk nucleosomes (Supplementary Fig. 12). We therefore suggest that the current view of an active promoter is in fact a composite of many different types of promoter nucleosome arrangements (Fig. 8a). Perhaps a single predominant nucleosome may be sufficient to establish the chromatin regulatory framework for a promoter. It is important

to add that while our promoter analysis was limited to a pre-determined number of promoter classes, it is conceivable that many other types of promoter chromatin arrangements exist that differ in more subtle ways. Perhaps it can be presumed that every single promoter is unique in its own way.

Our data suggests that changes in nucleosome accessibility rather than positioning play a more important role in regulating transcription when the phenotype of the cell changes. This indicates that nucleosome positioning and accessibility are not coupled, which confirms and extends previous studies[23]. We show that an unequivocal feature that defines an active promoter is a specifically positioned MNase resistant (total or H2A.Z) nucleosome either at the −2, −1, +1 or +2 positions. Conversely, inactive promoters appear to be more disordered lacking positioned nucleosomes which are also more MNase sensitive.

The nucleosome at the +1 position plays a particularly important role because it can affect the passage of RNA Pol II. Paradoxically, we reveal that an active promoter may contain a MNase-resistant H2A.Z nucleosome at the +1 position. It is attractive to speculate that even on an active promoter, this nucleosome may provide an important regulatory step to dampen and thereby regulate the level of transcription. This notion is consistent with the findings that in vitro[50], a H2A.Z+1 nucleosome greatly lengthens RNA Pol II crossing time and in Drosophila S2 cells, this H2A.Z nucleosome is anticorrelated with nucleosome turnover[39]. Further, H2A.Z intrinsically forms a stable histone octamer[42,51]. Intriguingly, the presence of a less well positioned, and more MNase sensitive H2A.Z nucleosome centred around the +1 position is correlated with strong gene repression (Fig. 3).

Whether a nucleosome occupies the TSS has been a matter of debate[1]. By analysing MNase sensitivity we identified an important role for H2A.Z at the TSS that simply would not be detected in nucleosome occupancy maps. Consequently, we show here that not only a H2A.Z nucleosome occupies the TSS, but it can be in either a MNase sensitive or resistant structure suggesting that H2A.Z has an important role in regulating TSS access. Available evidence indicates that the formation of a MNase-sensitive nucleosome reflects some DNA unwrapping, enabling subsequent non-histone protein binding without complete histone displacement[23,28,30,52]. Different mechanisms have been reported that can specifically destabilise a H2A.Z-containing nucleosome. For example, the underlying DNA sequence[53], the exchange of H3 with H3.3[41,54], histone acetylation[55] and the replacement of one copy of H2A.Z with H2A to form a heterotypic nucleosome[38,56]. Indeed, previous studies have shown that heterotypic H2A.Z-H2A nucleosomes[35], and nucleosomes containing both H2A.Z and H3.3 occupy an active TSS[25]. Therefore, it is attractive to speculate the TSS H2A.Z MNase-sensitive nucleosome observed here may in fact be a H3.3-heterotypic H2A.Z-containing nucleosome[10]. On the other hand, a H2A.Z MRN might be homotypic H2A.Z-H2A.Z nucleosome that lacks H3.3. Future experiments will explore this and other possibilities.

A third type of TSS chromatin structure was identified in this study where no nucleosome occupies the TSS i.e. nucleosome free. This TSS architecture correlates with a level of gene expression that is intermediate between a H2A.Z resistant and sensitive nucleosome. During the cell cycle, H2A.Z is depleted from promoters at mitosis and therefore this could explain why a proportion of active promoters lack H2A.Z at the TSS[35].

Finally, our data suggests that H2A.Z may have another important role at promoters by preventing promiscuous DNA access at the TSS and transcription factor binding sites because when H2A.Z is depleted in MCF-10A cells, there is a dramatic increase in subnucleosomal-sized particles at these important promoter DNA elements. The transition of nucleosomes into

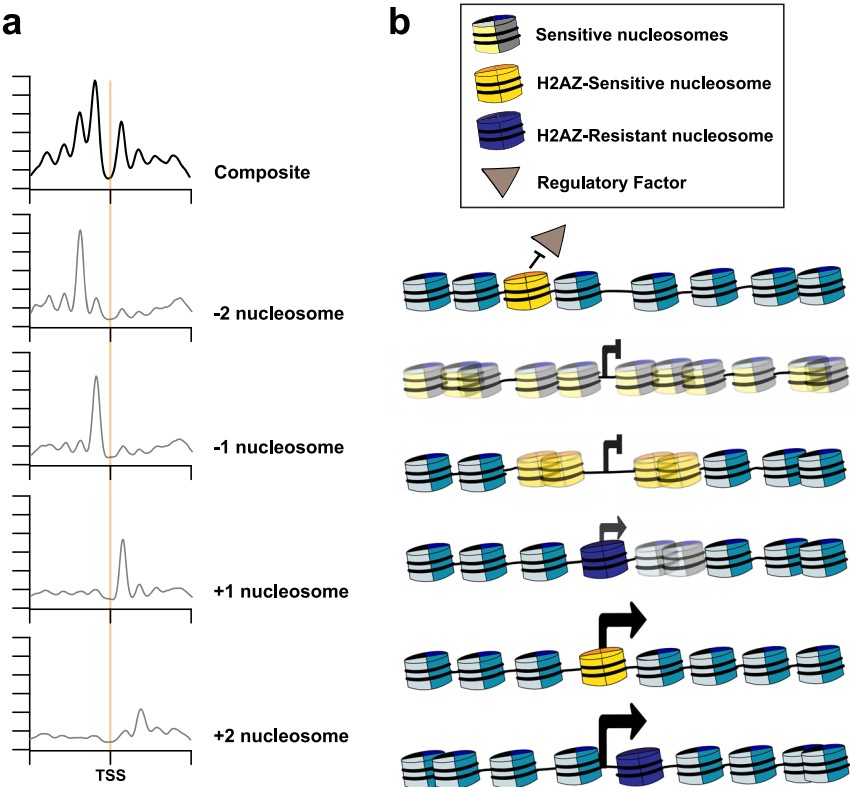

**Fig. 8 A model depicting how H2A.Z regulates promoter chromatin architecture. a** The current model for the arrangement of nucleosomes at a promoter is a composite of many different types of promoters with non-identical nucleosome occupancies at the −2, −1, +1 and +2 positions. **b** Multiple functions of H2A.Z at a promoter. H2A.Z may occupy transcription factor binding sites to prevent promiscuous transcription factor binding. Strongly repressed promoters are associated with poorly positioned micrococcal sensitive nucleosomes across the promoter or poorly positioned H2A.Z micrococcal sensitive nucleosomes at the −1 and +1 positions. Moderately repressed promoters contain a H2A.Z micrococcal resistant nucleosome at the TSS. Active promoters contain a micrococcal sensitive nucleosome at the TSS or a micrococcal resistant nucleosome at the +1 position with the latter chromatin structure displaying the highest level of gene expression.

subnucleosomal particles, critical for the subsequent invasion by non-histone proteins, was first shown to occur during spermatogenesis involving the testis specific histone variant H2A.L.2[57]. An important question to address in the future is to determine whether the loss of H2A.Z-specific chaperones (*YL1* or *ANP32E*) would also produce sub-nucleosomal DNA fragments at promoters[58].

The nature of the sub-nucleosomal sized particles observed here is unknown but could be nucleosomal hexamers or tetramers and/or reflect the binding of non-histone proteins. Significantly, sub-nucleosomal sized particles form at transcription factor binding sites only when their matched transcription factor is expressed suggesting that these small particles are transcription factors as indicated previously[28,30]. Another possibility, however, that cannot be excluded is that the loss of H2A.Z may allow other histone variants e.g. CENPA to be incorporated into promoters, which is known to wrap less DNA[59]. Future proteomic studies will elucidate the nature of these subnucleosomal fragments.

In conclusion, the results of this investigation suggest that H2A.Z has multiple roles in regulating promoter chromatin architecture and transcription dependent upon where it is located, and whether it exists in a stable or unstable nucleosomal form (Fig. 8b). This may provide an explanation as to why, despite an intensive effort, a single unifying role of H2A.Z in regulating transcription could not be identified.

## Methods

**Cell culture and lentiviral transduction.** MCF-10A and MCF-10Ca1a cell lines were grown in DMEM/Nutrient F12 (DMEM/F12) media supplemented with 5%

Horse Serum, 14 mM NaHCO3, 10 μg/mL insulin, 2 mM L-glutamine, 20 ng/mL human EGF, 500 ng/mL Hydrocortisone and 100 ng/mL Cholera Toxin. MCF 10A cells were obtained from the ATCC (CRL-10317) and MCF10Ca1a were obtained from the Barbara Ann Karmanos Cancer Institute (Detroit, Michigan). For EMT induction, MCF-10A cells were treated with 5 ng/ml TGF-β1 (Redsystems) for 8 days. MCF-10A cells were transduced with the lentiviral vector pLVTHM shH2A. Z[37]. GFP-positive cells were sorted 2 days post-transduction and further cultured for 8 days before being processed. RNA was purified using the RNeasy Mini Kit (Qiagen) and cDNA synthesised using the QuantiTect Reverse Transcription Kit (Qiagen). The knockdown of H2A.Z expression was analysed by qPCR on the 7900HT Fast Real-Time PCR system using SYBR Green I master mix (applied biosystems) and 0.15 mM primers[37]. Relative expression values were normalised to pooled housekeeping genes Glyceraldehyde 3-phosphate dehydrogenase (GAPDH), Beta-actin (B- actin), Ribosomal Protein S23 (RPS23) and Splicing Factor 3a Subunit 1 (SF3A1)[60]. Primer sequences are shown in Supplementary Table 1.

**RNA-Seq library preparation, sequencing, data processing and analysis.** All RNA-Seq experiments were performed in triplicate. Total RNA was isolated using the Qiagen RNeasy kit following manufacturer's instructions. Stranded mRNAseq libraries were constructed according to Illumina TruSeq stranded mRNA protocol with poly-A enrichment (Illumina). Libraries were sequenced with 2 × 75 bp paired-end on Illumina NextSeq500 instrument in high-output mode at Biomolecular Research Facility of the Australian Cancer Research Foundation, Canberra, Australia. Base-calling was performed using the "bcl2fastq2" script (V2.16) of the CASAVA pipeline. Technical aspects of the gene-level differential expression analysis were performed[37]. The UCSC hg19 "knownGene" set of transcripts were used as reference transcriptome against which transcript abundance estimation was performed, followed by collapsing of transcript abundances on gene level. Detailed analysis scripts implemented in "snakemake" and R scripts can be found at: https://github.com/JCSMR-Tremethick-Lab/MCF10APromoterAnalysis[66].

**Cell harvest and nuclei purification.** In all, 2.5 × 107 cells were harvested at, cross-linked in 1% formaldehyde in PBS, and incubated for 10 min at room

temperature. After the 10 min incubation, the cross-linking reaction was stopped by addition of 125 mM glycine. Next, the nuclei were isolated in nucleus isolation buffer containing: 10 mM HEPES at pH 7.8, 2 mM MgOAc$_2$, 0.3 M sucrose, 1 mM CaCl$_2$ and 1% Nonidet P-40. The nuclei were then pelleted by centrifugation at $1000 \times g$ for 5 min at 4 °C.

**MNase cleavage and purification of mononucleosomal and subnucleosomal DNA**. All experiments were performed as biological replicates. Nuclei were digested under either Light MNase conditions (40 gel units for 15 min at 37 °C) or Heavy MNase conditions (40 gel units for 30 min, then another 40 gel units for 30–40 min at 37 °C) (NEB M0247S) in MNase cleavage buffer (4 mM MgCl$_2$, 5 mM KCl, 50 mM Tris-Cl (pH 7.4), 1 mM CaCl$_2$, 12.5% glycerol). The MNase digestion reactions were stopped with 50 mM EDTA. Next, the protein-DNA crosslinks were reversed by treating the MNase-digested nuclei with 0.2 mg/mL proteinase K and 1% sodium dodecyl sulfate, and incubating overnight at 60 °C. The samples were then run and the nucleosomal ladder was separated on a 2% agarose gel. Following the separation of the DNA fragments, mononucleosome-sized and subnucleosomal-sized fragments (<200 bp) were isolated from the agarose gel. Next, the mono-nucleosomal- and subnucleosomal-sized fragments for all MNase conditions were combined for each respective sample. Following the combination of all fragments per sample, DNA was extracted with phenol-chloroform and precipitated with alcohol for 10 min at −20 °C. The DNA was then pelleted by centrifugation at $3000 \times g$ for 10 min at 4 °C, and dissolved in TE (0.1 mM EDTA, 10 mM Tris-Cl at pH 8.0).

**H2A.Z ChIP and western assays**. H2A.Z ChIP assays employing either low or high MNase-digested chromatin and the preparation of ChIP-Seq libraries for conventional genome-wide ChIP-Seq were carried out using our in house H2A.Z antibody, as extensively used previously[35,61–63]. H2A.Z ChIPs were performed on the same MNase-digested input chromatin. Western blots were performed as described[37]. A 1/1000 dilution was used for the anti-H2A.Z antibody. A 1/10,000 dilution was used for the anti-γ-tubulin antibody (Sigma-Aldrich, T6557). An uncropped western blot is shown in Source Data File 1.

**Mononucleosomal and subnucleosomal DNA library preparation**. Using the NEBNext® Ultra™ DNA Library Prep Kit for Illumina® (NEB #E7370S/L), DNA sequencing libraries were prepared for the mononucleosomal-sized and subnucleosomal-sized fragments for each sample. DNA was end-prepped using NEB Prep enzyme mix, end-repair reaction buffer (10X), and 30 ng of DNA for each sample, then held at 30 °C for 30 min and then at 65 °C for 30 min. Adaptors were ligated onto the end-repaired samples by adding NEB Blunt/TA Ligase Master Mix, NEBNext Adaptor for Illumina, and Ligation Enhancer and incubating at 20 °C for 15 min. The adaptor-ligated DNA was cleaned up using AMPure XP beads to remove any unwanted ligated products. The universal and indexed sequences were added by PCR using 23 μl of adaptor-ligated DNA fragments, NEBNext High Fidelity 2X PCR Master Mix, index primers provided in NEBNext Multiplex (NEB #E7335, #E7500) Oligos for Illumina, and Universal PCR Primers provided in NEBNext Multiplex (NEB #E7335, #E7500) Oligos for Illumina. Then PCR was done for 8 cycles (not including the initial denaturation and final extension). The adaptor-ligated DNA was cleaned up using AMPure XP beads to remove any unwanted products. The libraries were quality-checked using Agilent 2100 Bioanalyzer High-Sensitivity. Across the libraries, the samples ranged between 200 and 400 bp and there were no adaptor or primer dimers.

**Solution-based sequence-capture of DNA fragments within 2 kb of all human TSSs**. Previously, we combined MNase-seq with in-solution targeted enrichment of 2 kb surrounding TSSs of 21,857 human genes[9,13], as curated by NCBI RefSeq[44]. We termed this approach mTSS-seq. Size selected fragments (~50–200 bp) were used to prepare Illumina sequencing libraries and subjected to targeted enrichment utilising the custom-designed Roche Nimblegen SeqCap EZ Library. DNA fragments were captured according to the Roche Nimblegen protocol. Paired-end reads (see below) were aligned to the hg19 genome assembly (IHGSC 2001) and read densities were visually inspected with the UCSC genome browser. Globally, each data set contained an average of 85.7% of TSS-seq reads uniquely aligned to the genome (~10.4 million of uniquely mappable reads per experiment). Whole genome mapping data sets contain an average of 77.4% of TSS-seq reads in the absence of sequence capture. Of the reads that uniquely aligned to the genome, >98% overlapped with the targeted TSS regions compared to 2% in the absence of sequence capture.

**Performance of solution-based sequence capture**. We compared the TSS region coverage of the TSS-Seq libraries with the same libraries sequenced without TSS capture for three conditions: subsampling 5 M and 10 M reads, and including the fully sequenced libraries (genome-wide, non-captured libraries sequenced to approximately 277 M PE reads; details for TSS-Seq libraries provided Supplementary Fig. 1d). To simulate the low-coverage experiments by subsampling (using sambamba view -s) each included library to 5 million and 10 million PE reads and also included the fully sequenced libraries (normalised to input). The resulting BAM files were further processed using deeptools (3.2.1) to first calculate the

coverage in RPKM across known promoters (1000 bp ± of TSS) and then visualised the coverage heatmaps (Supplementary Fig. 1a–c). In order to illustrate the enrichment effect, we calculated the mean observed coverage in each of the libraries.

**Illumina paired-end sequencing and analysis**. Using a PE-50 single lane on an Illumina HiSeq 2500, HiSeq Flow Cell v3, the samples were loaded at 12 pM. The libraries were sequenced using standard Illumina sequencing protocols. Two kits were used: the TruSeq SBS Kit v3 and the TruSeq PE Cluster Kit v3 -cBot – HS. The reads were demultiplexed using the Casava Software (v1.8.4), and the library adaptors were removed using the cutadapt software. The sequenced fragments were aligned to the HG19 assembly of the human genome using bowtie2 2.1.0. Using samtools, non-unique and non-paired fragments were removed from the sequenced fragments. Nucleosome occupancy profiles were obtained by calculating the fragments per million that mapped at each base-pair in the SeqCap regions (bedtoolsCoverage). Midpoints ±30 bp (i.e. 60 bp centred on the nucleosome dyad) were used to plot nucleosome distributions and were determined through the calculation of centre fragments in 60 bp windows at a 10 bp step in the 2 kb surrounding each TSS. Further analysis of the nucleosome distributions was done in the R environment, R 2.15.1, using our lab-developed software, genmat (https://github.com/dvera/genmat). See Supplementary Table 2 for sequencing information for each sample.

**Clustering analysis and alluvial plots**. Analysis of nucleosome distributions was done in the R environment using our custom tools (https://github.com/dvera, packages travis, genmat, gyro). matHeatmap was used to plot normalised rpm mTSS-seq data for all RefSeq genes ±2 kb surrounding the annotated TSSs. Average plots represent the average value of all promoters in a respective cluster in 10 bp windows across the 2 kb SeqCap region. K-means clustering of nucleosomal data was performed using the matHeatmap function, using the Hartigan and Wong algorithm. Dinucleotide periodicity was calculated from total 147–150 bp nucleosomal fragments using the bedWords function. Gene ontology analyses based on heatmap gene classifications were performed using two unranked sets of genes, the target set of genes and the total mTSS-seq gene list as background[65]. Alluvial diagrams were created using the R package "alluvial" (Bojanowski M and Edwards R (2016). *alluvial: R Package for Creating Alluvial Diagrams*. R package version: 0.1–2, https://github.com/mbojan/alluvial), based on the results of the k-means clustering analyses.

**Sequence analysis of subnucleosomal peak DNA**. In order to determine if specific DNA motifs, which could represent transcription factor binding sites (TFBS), are associated with the observed subnucleosomal fragments, and if furthermore the subnucleosomal fragments originate from known H2A.Z peak sites, we performed our analysis in following steps: First, we used the aligned sub-nucleosomal fragments as input for peak calling using MACS2 v.2.1.2-0. MACS2 was run without control/input libraries, as these fragments are the result of an MNase digest of DNA without immuno-precipitation. Instead the local background distribution ("lambda") was imputed from the treatment/MNase libraries. The genome size parameter was adjusted to 40,999,507 which represents the total number of nucleotides covered under the promoter capture array, and thus represents the maximum mappable genome. For H2A.Z containing nucleosomes, MACS2 peak calling was performed with independently for each biological replicate H2A.Z ChIP library. MNase-digested DNA libraries were pooled and used as the control. The q-value cut-off parameter was set to 0.9 in order to increase the peak list. The two replicate peak lists were then used to estimate irreproducibility of discovery rate (IDR) of each peak, and a cut-off of IDR 0.1 was chosen to select the most reproducible set of peaks for further analysis. The genome size parameter was adjusted to 40,999,507, as explained above.

A "snakemake" implementation of the peak calling pipeline can be found at https://github.com/JCSMR-TremethickLab/MCF10APromoterAnalysis/blob/master/snakemake/rules/shortFragments_macs2_peak_calling.py[66].

The lists of subnucleosomal fragment peaks and summits were loaded into R/Bioconductor using functions from the "data.table" package. We chose the summits of peaks with a q-value < 0.1 and a fold-enrichment of >4 as the central point of subnucleosomal peaks. For each summit passing the filter threshold, we exported 500 bp of DNA sequence surrounding the summit site (single nucleotide) as FASTA files using the BSgenome package (https://bioconductor.riken.jp/packages/3.4/bioc/manuals/BSgenome/man/BSgenome.pdf).

The FASTA files were used as input for "meme" and motif discovery was performed using the objective functions "centrally enriched" as well as "centrally distributed". Minimum word size was set to 8, maximum to 15. The background hidden markov model was estimated from the actual promoter sequences, as captured by our promoter capture array using the "fasta-get-markov" utility supplied as part of the MEME suite (v5.0.5). E-value threshold for motif discovery was set to 0.05 and mode was set to "zoops".

The results from the "meme" search were used as input for the "tomtom" search in the "HOCOMOCO v11 full HUMAN" database of human transcription factor binding sites. The "snakemake" script driving this abridged MEME analysis pipeline can be found at https://github.com/JCSMR-Tremethick-Lab/

MCF10APromoterAnalysis/blob/master/snakemake/rules/
shortFragments_meme_processing.py[66].

Both "meme" and "tomtom" search result files were loaded into R using the fread function from the "data.table" package and further analysis was performed using the information from peak calling and differential gene expression analysis. All scripts in the integrative analysis can be found at https://github.com/JCSMR-Tremethick-Lab/MCF10APromoterAnalysis/tree/master/R/TFBSAnalysis[66].

**Analysis of publicly available histone PTM ChIP-Seq data.** All software packages were installed using the pythonic package manager "conda" through the "bioconda" channel[67]. Publicly available ChIP-Seq data for MCF10A cells were obtained from NCBI's Sequence Read Archive using "sra-tools" (2.10.1, https://github.com/ncbi/sra-tools) downloaded to our local HPC system using "prefetch" and exported in FASTQ format using "fastq-dump". Raw FASTQ sequencing data for each library were processed with fastp version 0.19.5[68] in order to remove potential adaptor sequences, trim low-quality bases and to perform quality control. The trimmed and quality-checked reads were then aligned to the unmasked human reference genome build GRCh37/hg19 (UCSC annotation) using bowtie2 2.3.5[69] with minor modifications to the default parameters (''—no-mixed –no-discordant''), and allowing for a maximum insert size of 500 bp. The resulting SAM output was further processed using samtools version 1.9[70] and picardTools version 2.20.1 (Picard toolkit, 2019). In a first step, alignments with a quality score <10 were removed, followed by marking and removal of duplicate reads using picardTools 'MarkDuplicates'. The sorted, quality filtered and de-duplicated BAM files were than indexed using samtools 'index' and insert size distributions estimated with picardTools 'CollectInsertSizeMetrics'. Aligned ChIP-Seq and input data were further inspected using a set of quality control tools provided as part of the 'deepTools'[71] Pearson correlation of read coverages were calculated and used to estimate the technical variability between replicates. DeepTools was further used to create bigwig files (using bamCoverage), and these bigwig files were used for calculating enrichment scores over input coverage (bigwigCompare) which were then visualised for selected genomic regions using computeMatrix and plotHeatmap/plotProfile. We used the set of genes defined by our k-means clustering for plotting coverage for the individual histone marks in the promoter regions as defined by our TSS arrays. A snakemake workflow implementing this pipeline can be accessed at https://github.com/skurscheid/mcf10_promoter_profiling[72].

**Reporting summary**. Further information on research design is available in the Nature Research Reporting Summary linked to this article.

## Data availability

The data discussed in this publication have been deposited in NCBI's Gene Expression Omnibus[64]. The parent Super Series containing all data is accessible through GEO Series accession number "GSE134299", the MNase-seq and ChIP-seq data are available through GEO Series "GSE134297]", and the RNA-seq data is accessible through GEO Series "GSE134298". All other relevant data supporting the key findings of this study are available within the article and its Supplementary Information files or from the corresponding author upon reasonable request. A reporting summary for this Article is available as a Supplementary Information file. Source data are provided with this paper.

## Code availability

Code is available through:

https://github.com/JCSMR-Tremethick-Lab/MCF10APromoterAnalysis[66].

https://github.com/JCSMR-TremethickLab/MCF10APromoterAnalysis/blob/master/snakemake/rules/shortFragments_macs2_peak_calling.py[66].

https://github.com/JCSMR-Tremethick-Lab/MCF10APromoterAnalysis/blob/master/snakemake/rules/shortFragments_meme_processing.py[66].

https://github.com/JCSMR-Tremethick-Lab/MCF10APromoterAnalysis/tree/master/R/TFBSAnalysis[66].

https://github.com/skurscheid/mcf10_promoter_profiling.

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

## Acknowledgements

We acknowledge the excellent high-throughput sequencing service provided by the JCSMR Biomolecular Research Service headed by Stephanie Palmer. We thank Jen Kennedy for critical reading of the manuscript, Emily Darrow for productive discussions regarding the analysis and Danielle Cole for assistance with Illustrator. This work was supported by grants from the National Institutes of Health, R01 DA033775 (J.H.D.); The FSU Foundation Developing Scholar Award, DSA040358 (J.H.D.); and the National Health and Medical Research Council, 1104340 (D.J.T.).

## Author contributions

L.C. and S.K devised and performed the bioinformatic analyses. M.N. and R.D. did the wet-lab experiments. D.L.V. wrote the pipelines for data analysis for the Dennis Lab. J.H.D. and D.J.T. designed and supervised the study. L.C., J.H.D. and D.J.T. interpreted the data and wrote the manuscript.

## Competing interests

The authors declare no competing interests.
