## [Peer Review File · Nature Communications]

Reviewers' comments:

Reviewer #1 (Remarks to the Author):

This paper is focused on the chromatin organization of active and inactive promoters. The genome-wide positioning and accessibility to micrococcal nuclease (MNase) of both conventional and H2A.Z nucleosomes at promoter regions were analyzed. The data showed that, in contrast to the current view, the promoters displayed multiple nucleosome arrangements. Interestingly, this arrangement was found not to depend on the cellular states. In addition, a number of distinct types of MNase digestion profiles that do not change with the cellular types were detected. Active promoter displayed H2A.Z-dependent MNase accessibility and +1 nucleosome resistance. In agreement, MNase digestion of H2A.Z depleted nuclei generates subnucleosomal DNA sized fragments at the promoter transcription factor binding sites.

This is an excellent manuscript. The described experiments are elegant and professionally carried out. The manuscript is very well presented and, despite of the complexity of the subject, it is easy to read and understand.

I have no major critical notes. Instead, I would like to propose some issues related to the nucleosome promoter organization to be considered for the Discussion chapter of the manuscript. Some additional bioinformatic analysis could also be done. My feeling is that this would strengthen the manuscript.

1. Different groups of promoters exhibited distinct, H2A.Z-independent nucleosome arrangements. I wonder whether this promoter property would not depend on the underlying promoter nucleosome DNA sequence. Would it be possible to study this by analyzing the nucleosome-associated DNA sequences in the distinct groups of promoters?

2. The digestion with MNase in H2A.Z-depleted nuclei resulted in the generation of < 125 bp subnucleosomal fragments (SFs, < 125 bp) at the transcription binding sites. Why such fragments are generated? Do they reflect some types of less stable nucleosomes or either partially (hexasomes) or totally (tetrasomes) H2A-H2B dimer depleted nucleosomes? Some in vitro MNase digestion experiments using reconstituted hexasomes/tetrasomes particles could be done and the digestion profile compared with the SF profiles of MNase digested nuclei. Alternatively, the SF profiles could be compared with some available MNase nucleosome digestion data, which might be sufficient to shed some light of the raised issue.

3. Is it possible the increased accessibility to MNase in the H2A.Z-depleted nuclei to reflect the presence of some histone variant nucleosomes with lower stability, assembled "de novo" on the promoters as a result of H2A.Z depletion? For example, both H2A.Bbd and CENP-A histone variant nucleosomes are known to be less stable and both histone variants appeared to be associated with promoter regions. This should be discussed.

4. I wonder if the absence of the H2A.Z histone chaperones (YL1and/or ANP32E) would generate the SF particles upon Mnase digestion. Indeed, the absence of these chaperones would lead to genome-wide perturbations of the H2A.Z nucleosome pattern and thus, one may expect no generation of promoter resistant particles. This could be addressed in the Discussion.

5. The authors have shown in the past that H2A.Z is depleted from promoters at the onset of mitosis. Then, no MNase more resistant nucleosome should be observed at the promoters region at mitosis. This could be noted in the Discussion.

Minor notes

1. My feeling is that the Abstract does not clearly summarize the manuscript data, i.e. it is very "general". It should be re-written and more details should be included.

2. H2A.Z could not be, as stated in the text, “activator/repressor” of transcription, but associated with such events. To be corrected.

3. The presence of subnucleosomal particles (SFs) and their origin were analyzed in details during spermatogenesis. The corresponding papers should be cited.

Reviewer #2 (Remarks to the Author):

In this study, the authors analyzed the nucleosome organization in cells including MCF-10A with or without treatment with TGF- β , MCF-10CA1a, MCF-10A with shH2A.Z. The authors claimed that they have generated a high-resolution profile of total nucleosomes and H2A.Z nucleosomes around TSS. Different from previous studies, the authors found that there existed different types of nucleosome arrangements, which are conserved among the four cell types. On the other hand, they observed that there is a dramatic change of nucleosome sensitivity or resistance across the four cell types. Therefore, they concluded that nucleosome occupancy is decoupled from nucleosome accessibility.

The authors observed a dramatic increase of sub-nucleosome fragments around TSS upon the knock-down of H2A.Z. With the loss H2A.Z, H2A.Z nucleosome is destabilized, inducing an increase in MNase-accessibility. Next, the authors showed that, upon the loss of H2A.Z, the increase of sub-nucleosome fragments at TSS are correlated with the gene expression. Also, these TSS regions contain many transcription factor binding sites (TFBS).

Major comments:

The authors' findings about the existence of different conserved nucleosome patterns and the decoupling between nucleosome occupancy and nucleosome accessibility are potentially interesting. However, due to the limited sequencing depth of the libraries and limited analyses of the data, this conclusion appears to be too preliminary.

For other observations associated with the loss of H2A.Z, I have concerns about the novelty of the findings. Previous studies have shown that H2A.Z can control the nucleosome stability and mobility [1]. Thus, it is not surprising that regions with an increase in sub-nucleosome fragments are related to genes with high expression level and enriched with TFBS.

Overall, more evidence is need to support the major findings before the manuscript can be published in a prestigious journal such as Nature Communications.

Specific Points:

1) According to Table S3, the average number of reads for MNase-seq is in the order of 10 millions. Then the number of reads between 140-180bp may be about 5-7 millions. It is difficult to claim that the nucleosome profile is high-resolution with this low depth of sequencing. Importantly, the observation of different nucleosome arrangements can be just an artifact due to the low sequencing depth. The authors should obtain a sequencing depth comparable to this study (GSM1004654) in order to draw a meaningful conclusion.

2) If the existence of different nucleosome arrangements is true, it is not reasonable that this fundamental nucleosome organization only happens in MCF-10A cells. Such a phenomenon should also be observed in other cell types. A very easy way is that the authors can do the same analysis using the published MNase-seq data sets such as GSM1004654. If the same distinct nucleosome arrangements can be observed, it would make the conclusion much stronger.

3) Instead of selecting all 22000 genes, can the authors observe the phenomena if the clustering analysis is applied to the top 10% or top 5% of the promoter based on the gene expression level?

4) The authors did not provide some important information in the text about the data analyses. For example, how did the authors determine the number of clusters? How did they choose 7? Did

the author apply a gaussian smoothing at individual genes before plotting the nucleosome patterns? If not, the authors should try to do this since it may provide more stable and better averaged nucleosome profiles.

Reference:

1) Rudnizky, S., Bavly, A., Malik, O. et al. H2A.Z controls the stability and mobility of nucleosomes to regulate expression of the LH genes. *Nat Commun* 7, 12958 (2016).

Reviewer #3 (Remarks to the Author):

Cole and colleagues present in this manuscript a very interesting approach to generate high resolution maps of "stable" and "less stable" nucleosomes at gene promoters based on applying different conditions during MNase digestion. Additionally, they have done the same approach to interrogate the properties of H2A.Z in particular. With these data, they attempt to build a map of nucleosome positioning, nucleosome stability/accessibility and the presence of subnucleosomal fragments at gene promoters and overlay H2A.Z-nucleosomes in these maps. The overarching conclusion of this study is that the classic view of -1, NFR, +1 nucleosomes are active promoters is a composite of multiple different states of these promoters; they find that globally there are 7 promoter states based on differential nucleosomal properties. The most interesting aspect of this study is their findings in H2A.Z-specific nucleosomes, where several types of H2A.Z-nucleosomes are found in the context of gene expression.

This manuscript describes concepts that have been previously shown including the existence of an unstable H2A.Z nucleosome at the TSS of active genes PMID: 19633671; the generation of high-resolution maps that allow the detection of "unstable" nucleosomes PMID: 27889238, and PMID: 27151365; and the concept of that nucleosome positioning does not change but nucleosome accessibility does during transcription PMID: 27151365 and PMID: 28356342. Based on this, the level of novelty of this study is somehow challenged.

The most novel aspect for this reviewer is the combination of various NGS-based methods for the generation of complex nucleosome maps: a high-resolution method for promoter interrogation (mTSS-seq) combined with different variables: two conditions of MNase treatment to investigate the different "nature" of total nucleosomes and H2A.Z-nucleosomes (stables, positioned and sub-nucleosomal). The major flaw of this novel aspect is the way they have interpreted and integrated the data which does not lead to solid conclusions (I will give some examples of this fact below)

Major concerns:

1- The data presented seems quite preliminary, it provides useful information in several particularities of H2A.Z nucleosomes at promoters as a good starting point but it does not provide solid conclusions on the functionality of the different subclasses of H2A.Z-nucleosomes based on their MNase properties nor in the general nucleosomal mapping architecture at gene promoters. It is not a rounded-up story that deepens into the molecular mechanism of their observations. For example: the cellular models presented could have been better used to study nucleosome changes at promoters of genes that directly change upon TGFB treatment or H2A.Z knock down or during carcinogenesis; but the authors just comment that the promoters have the same clusters for nucleosome positioning and major changes in nucleosome accessibility, but these evidences are not put in the context of gene expression and or transcriptional changes.

2- The authors perform k mean clustering for each cell model and for each specific biological question, this generates clusters that look different despite that they come from the same dataset (Fig 1A versus 1C and 1B versus 1D), this creates lots of confusion for the readers very as it is nearly impossible to be able to see common or integrated patterns among cell models and promoter states, how do the clusters of nucleosome positioning (Fig1A and B) overlap o to which clusters of nucleosome accessibility correspond (Fig 1C and D). On top of this, the number of clusters seem randomly chosen, I recommend performing the elbow method to determine the optimal number of clusters.

3- For the changes in Fig 1C and D this reviewer asks how quantitative is the MNase mTSS-seq method and what "normalising methods or spike ins" have they used. The differences should be measurable, and statistics need to be applied.

4- The association of these nucleosome maps with gene expression and their dynamics when transcription is altered are very vague. The existence of "poised promoters" is not even mentioned in the whole manuscript and there isn't any clear conclusion of how the "nucleosome architectural map" for cell states, cancer state, transcriptional state or promoter state should look like. In this context, the title of this manuscript does not reflect the conclusions of the study

5-Figure 1C/D and S6. The authors described that H2A.Z-nucleosome constitutes the +1 MNR-nucleosomes however the fact that upon H2A.Z KD there is a clear change on nucleosome properties spanning +1kb/-1kb towards MNR is completely ignored, this observation needs to also be commented by the authors

6-The insights on the different types of H2A.Z-nucleosomes according to the promoter transcriptional state could have been much more explored to at least give a better definition of these nucleosomes: are they homotypic, heterotypic; do any of them they have H3.3? Is H2A.Z acetylated in any of these H2A.Z nucleosomes? Is there a difference in H2A.Z isoforms?

7- The plots from Fig 1D are very confusing to interpret and are quite misleading, the MRN give a negative value that it looks like there is a NFR instead of a nucleosome, the authors should overlay (all in positive values) the MSN and MRN data in the same plot.

8- The average gene expression represented as a straight line of a mean across the promoter is unacceptable (Fig. 2b, Fig3a and 3b), please use violin or box plot representations in a separate plot to see the deviation and include p-values.

9- H2A.Z nucleosomes and MNase total heatmaps look almost the same (Fig 1a/b and S4), could authors make a comment on this? Is this expected to occur in all +/- 1kb promoters? I would recommend to have extra controls of other histone variants H3.3 or histone PTMs that are expected to have a discrete localisation in the promoters. Have the in-house H2A.Z antibody been properly checked for its specificity for ChIP?

10- Overall, the manuscript and their figures have not been carefully written and made, respectively. The manuscript is very hard to follow, there are many missing labels or misleading labels in the figures (this gets quite bad in the supplementary figures), there is missing information in the material and methods (western blots for KD validation?). In general, the figures are not self-explanatory, the figure legends lack of explanation and clarification of the abbreviations and some terms are used differently when they are the same thing, in example total MNase-seq and mTSS-seq, there are missing references, etc.

11- Downsampling analyses for comparison between H2A.Z ChIP-seq and H2A.Z mTSS-seq. The authors show a clear improvement of 19X fold when they compare their mTSS-seq and standard H2A.Z ChIP-seq when the latter is down sampled to 2.5 PE million reads. I wonder how the mTSS-seq data performs when it is compared to a H2A.Z-ChIP with standard 45 PE million reads.

Minor comments:

1-The cell model systems need to be better validated; the authors should show the phenotypic and transcriptional effects of TGFB treatment and H2A.Z KD to demonstrate that the cells undergo EMT.

2- In the sentence "Unexpectedly, a distinct H2A.Z MSN occupies the TSS" the word

"unexpectedly" should be removed as this type of nucleosome has been already described PMID: 19633671.

3- Likewise, the word "unexpectedly" should be removed in the sentence "Unexpectedly, for both total (Fig. 1A) and H2A.Z-containing (Fig. 1B) nucleosomes, we observe different clusters that display a single, high occupancy nucleosomes located at different positions within 500 bp of the TSS."; a similar concept has been described previously PMID: 27889238.

4- The word "coverage" is wrongly used when it should be referred as "depth".

5- In Figure 3, please add expression fold change of the promoters that show increase or decrease subnucleosomal fragments.

General comments to reviewers

We thank all the reviewers for their constructive comments, which has made the manuscript considerably stronger. The major changes are: **1)** a substantial re-write of the manuscript (including changing the title to “ Multiple roles of H2A.Z in regulating promoter chromatin architecture” to better highlight the novel findings of the manuscript; **2)** extensive new analyses, including tracking changes in promoter chromatin architecture and gene expression following TGF β induction, malignant transformation and H2A.Z knockdown by generating alluvial plots and **3)** a new analysis of publicly available histone modification ChIP-Seq data for MCF10A cells that confirms our new classification of the different types of active and inactive promoter nucleosome arrangements based on our mTSS-seq approach. Minor changes include further analyses that has strengthened our finding that H2A.Z prevents promiscuous access of TF binding sites.

The key novel findings of the manuscript are that: **1)** H2A.Z has multiple roles in regulating promoter chromatin architecture, which is dependent upon where it is located in the promoter (upstream or downstream of the TSS, at the TSS or at TF binding sites), and whether it exists in a stable or unstable nucleosomal form (see new model in Fig. 8b); **2)** not only an unstable H2A.Z nucleosome occupies the TSS but a stable H2A.Z nucleosome can also sit there and these promoters exhibit lower levels of gene expression. This suggests that regulating the stability of a TSS H2A.Z-containing nucleosome may play an important role in controlling TSS accessibility; **3)** unexpectedly, an active promoter has a MNase stable H2A.Z nucleosome at the +1 position, which is correlated with major changes in gene expression when H2A.Z is removed; **4)** H2A.Z nucleosomes occupy transcription factor binding sites to prevent promiscuous access and finally **5)** we have uncovered a diversity of different types of active and inactive promoter structures that were previously unknown.

In conclusion, the role of H2A.Z in regulating transcription has been enigmatic. This is because the field has tried to uncover a unifying role for H2A.Z in regulating transcription. Our study has shown that such a unifying role does not exist because it can perform multiple and even opposing roles in a promoter.

Reviewers comments:

Reviewer #1 (Remarks to the Author):

1. Different groups of promoters exhibited distinct, H2A.Z-independent nucleosome arrangements. I wonder whether this promoter property would not depend on the underlying promoter nucleosome DNA sequence. Would it be possible to study this by analysing the nucleosome-associated DNA sequences in the distinct groups of promoters?

This is a great question and we are working on the development of the bioinformatic tools to tackle this complex problem. Therefore, we argue that this will be the basis of a future manuscript and is therefore beyond the scope of this manuscript. However, we have looked at the periodic occurrence of AA/TT/AT/TA dinucleotides for all bulk and H2A.Z nucleosomes (New Supp Fig. 11). The characteristic 10 bp periodicity of the AT dinucleotide that suggests rotational phasing is observed but interestingly, H2A.Z appears to be less dependent on this AT phasing signal compared to bulk nucleosomes.

2. The digestion with MNase in H2A.Z-depleted nuclei resulted in the generation of < 125 bp subnucleosomal fragments (SFs, < 125 bp) at the transcription binding sites. Why such fragments are

generated? Do they reflect some types of less stable nucleosomes or either partially (hexasomes) or totally (tetrasomes) H2A-H2B dimer depleted nucleosomes?

The nature of the sub-nucleosomal sized particles observed here is unknown but could be nucleosomal hexamers or tetramers and/or reflect the binding of non-histone proteins. Significantly however, subnucleosomal sized particles form at transcription factor binding sites only when their matched transcription factor is expressed suggesting that they do represent TF binding. Supporting this hypothesis, we have now included a new Figure 6 (panel B) that displays heat maps of < 125 bp and >125 bp MNase DNA fragments at c-Fos, c-Myc and STAT3 binding sites. These heat maps reveal sub-nucleosomal sized particles at the TF binding site flanked on either side by H2A.Z-containing nucleosomes.

Some in vitro MNase digestion experiments using reconstituted hexasomes/tetrasomes particles could be done and the digestion profile compared with the SF profiles of MNase digested nuclei. Alternatively, the SF profiles could be compared with some available MNase nucleosome digestion data, which might be sufficient to shed some light of the raised issue.

As discussed above, we have now compared SF profiles with MNase nucleosome digestion data (Figure 6b). Further, this interpretation is consistent with previous studies (e.g. Ref 28) that showed that TF binding resulted in the formation of sub-nucleosomal particles. Additional evidence that these sub-nucleosomal particles may be TFs is that these small particles are resistant to MNase digestion whereas hexasomes and tetrasomes are not since they are highly unstable.

3. Is it possible the increased accessibility to MNase in the H2A.Z-depleted nuclei to reflect the presence of some histone variant nucleosomes with lower stability, assembled “de novo” on the promoters as a result of H2A.Z depletion? For example, both H2A.Bbd and CENP-A histone variant nucleosomes are known to be less stable and both histone variants appeared to be associated with promoter regions. This should be discussed.

H2A.Bbd is testis specific and therefore not expressed in the cell lines used in this investigation. The interesting possibility that the loss of H2A.Z may allow CENP-A binding is now included in the discussion (First paragraph, page 19).

4. I wonder if the absence of the H2A.Z histone chaperones (YL1 and/or ANP32E) would generate the SF particles upon Mnase digestion. Indeed, the absence of these chaperones would lead to genome-wide perturbations of the H2A.Z nucleosome pattern and thus, one may expect no generation of promoter resistant particles. This could be addressed in the Discussion.

This interesting question is now proposed in the discussion (3rd last paragraph page 19).

5. The authors have shown in the past that H2A.Z is depleted from promoters at the onset of mitosis. Then, no MNase more resistant nucleosome should be observed at the promoters region at mitoses. This could be noted in the Discussion.

Thank you for reminding us of this important information. This is now included (4th last paragraph page 19).

Minor notes

1. My feeling is that the Abstract does not clearly summarize the manuscript data, i.e. it is very “general”. It should be re-written and more details should be included.

You are absolutely correct. We have re-written part of the abstract and indeed, changed the title of the manuscript to better explain the novel findings of this study.

2. H2A.Z could not be, as stated in the text, “activator/repressor” of transcription, but associated with such events. To be corrected.

We apologise for making such a fundamental mistake, which is now corrected.

3. The presence of subnucleosomal particles (SFs) and their origin were analyzed in details during spermatogenesis. The corresponding papers should be cited.

We totally agree and have included this description of the formation of sub-nucleosomal fragments during spermatogenesis (last paragraph page 18, Ref 54).

Reviewer #2 (Remarks to the Author):

Major comments:

The authors’ findings about the existence of different conserved nucleosome patterns and the de-coupling between nucleosome occupancy and nucleosome accessibility are potentially interesting. However, due to the limited sequencing depth of the libraries and limited analyses of the data, this conclusion appears to be too preliminary.

We apologise for obviously not explaining properly that we used a MNase-Transcription Start Site Sequence Capture approach whereby we targeted for the enrichment of 2 kb surrounding TSSs of 21,857 human protein-coding genes. This increases the sequencing depth and resolution by reducing the size of the genome for sequencing from 3.4Gb to 40Mb. Moreover, in Figure S1 we demonstrated that compared to standard CHIP-Seq, we obtained around a 18 fold enrichment across all promoters. In other words, while the total number of sequencing reads appear to be low, they are targeted to the promoter. Therefore, we argue that this manuscript is new because it represents the first high-resolution H2A.Z-containing, MNase stability, nucleosome maps for all human promoters in different normal and abnormal cellular states. We have rectified this misunderstanding by including a new section at the beginning of the results explaining our MNase-Transcription Start Site Sequence Capture approach.

For other observations associated with the loss of H2A.Z, I have concerns about the novelty of the findings. Previous studies have shown that H2A.Z can control the nucleosome stability and mobility [1]. Thus, it is not surprising that regions with an increase in sub-nucleosome fragments are related to genes with high expression level and enriched with TFBS.

Concerning novelty, it is clear that we did not describe the new observations in the previous manuscript particularly well, so we have undertaken major revisions and indeed changed the title to “Multiple roles of H2A.Z in regulating promoter chromatin architecture”. What is novel (as described above) is that we have found that these multiple roles of H2A.Z in regulating promoter chromatin architecture is dependent upon where it is located in the promoter (upstream or downstream of the TSS, at the TSS or at TF binding sites), and whether it exists in a stable or unstable nucleosomal form (see new model in Fig. 8b). Further, the role of H2A.Z in regulating transcription has been enigmatic. This is because the field has tried to

uncover a unifying role for H2A.Z in regulating transcription. Our study has shown that such a unifying role does not exist because it can perform multiple and even opposing roles in a promoter.

We disagree with the statement that “it is not surprising that regions with an increase in sub-nucleosome fragments are related to genes with high expression level and enriched with TFBS”. Most studies have focused on H2A.Z being at either at the promoter +1 or -1 position. We show here that, in cells, the loss of H2A.Z causes a dramatic increase in overall promoter accessibility resulting in a dramatic increase the formation of sub-nucleosomal particles at the TSS and at TF binding sites. In other words, H2A.Z nucleosomes are sitting on or near TF binding sites to prevent promiscuous promoter access. This observation was unexpected. We have added more data (Fig. 6 and 7) and described in more detail this major observation in the last paragraph of the Results section. Further, this manuscript shows that the current view of an active promoter is in fact a composite average of many different types of promoter configurations.

Concerning the study by Rudnizky et al., (which we have now included in the Discussion), it examined only two promoters (and two H2A.Z nucleosomes on each promoter) whereas our global study examined over 19,000 H2A.Z nucleosomes. Further, the study by Rudnizky et al., showed that H2A.Z weakens histones interactions with DNA. In contrast, our study shows that H2A.Z can both weaken and strengthen histone-DNA interactions (as assayed by MNase sensitivity) with the majority of H2A.Z nucleosomes displaying stronger histone-DNA interactions. This highlights the importance and significance of our study because we analysed all human promoters and not just a few. Further, in contrast to the current view, a H2A.Z MNase resistant (and not sensitive) nucleosome occupies the important +1 position on an active promoter. Finally, while there is a wealth of *in vitro* data (including our own work), this study is the first to describe the physical/MNase properties of all H2A.Z-containing nucleosomes and how they change in different physiological states.

Specific Points:

1) According to Table S3, the average number of reads for MNase-seq is in the order of 10 million. Then the number of reads between 140-180bp may be about 5-7 million. It is difficult to claim that the nucleosome profile is high-resolution with this low depth of sequencing. Importantly, the observation of different nucleosome arrangements can be just an artifact due to the low sequencing depth. The authors should obtain a sequencing depth comparable to this study (GSM1004654) in order to draw a meaningful conclusion.

As described above, we sequenced captured only the promoter regions thus indeed this analysis is a high-resolution analysis (Supplementary Figure 1). Moreover, a low sequencing depth would produce only the two promoter clusters (the existing model of promoter nucleosome organisation) and not the variety of different promoter classes that we report in this manuscript.

2) If the existence of different nucleosome arrangements is true, it is not reasonable that this fundamental nucleosome organization only happens in MCF-10A cells. Such a phenomenon should also be observed in other cell types. A very easy way is that the authors can do the same analysis using the published MNase-seq data sets such as GSM1004654. If the same distinct nucleosome arrangements can be observed, it would make the conclusion much stronger.

Our manuscript did show the same nucleosome sensitive arrangements in different cell lines. We repeated the mTSS-Seq approach using cell lines completely different from epithelial MCF-10A cells; the B-lymphoblastoid and lung fibroblast cell lines GM12878 and IMR90, respectively (Supplementary Fig.

7). This showed that the different MSN and MRN promoter types observed in MCF-10A cells are recapitulated in these different cell lines. Further, it is important to point out that in our study, we compared MCF-10A cells (an epithelial cell line) with MCF-10A cells treated with TGF β (where a dramatic phenotypic change occurs where the epithelial state is converted to a mesenchymal state). Further, in our study we also examined the highly malignant cell line (MCF-10CA1a cells). In other words, all three cell lines are completely different from each other yet they show similar promoter organizations. Taken together, we can argue strongly that the nucleosome arrangements that we observe are not unique to MCF-10A cells.

3) Instead of selecting all 22000 genes, can the authors observe the phenomena if the clustering analysis is applied to the top 10% or top 5% of the promoter based on the gene expression level?

In Figure 2A and Supplementary Fig. 8, we did do this. We took the top 25% of expressing genes and sorted them based on total and H2A.Z nucleosome occupancy (that showed strong nucleosome positioning for active genes), and total and H2A.Z nucleosome MNase sensitivity (that showed a H2A.Z MNase sensitive nucleosome at the TSS and a H2A.Z MNase resistant nucleosome at the +1 position). Importantly, we independently sorted these four profiles (Fig. 2A) into seven clusters (Supplementary Fig. 8) and these clustering analyses recapitulates the clusters observed in Figure 1. In addition, in Supplementary Fig. 6, the top sextile of expressed genes in MCF10A cells were sorted on maximum occupancy of the +1 nucleosome.

4) The authors did not provide some important information in the text about the data analyses. For example, how did the authors determine the number of clusters? How did they choose 7?

Our choice of seven was based on the empirical observation that seven clusters was the minimum number of clusters that generated robust promoter categories without producing redundant categories. In other words, clusters greater than seven did not reveal any new promoter configuration that was not observed when $k=7$ (see point 2, reviewer 3 below). This is now explained more clearly in the text. It is important to note that we added the following to the discussion “ that while our promoter analysis was limited to a predetermined number of promoter classes, it is conceivable that many other types of promoter chromatin arrangements exist. In fact, it is likely that no two promoter chromatin organizations are the same.”

50 Did the author apply a gaussian smoothing at individual genes before plotting the nucleosome patterns? If not, the authors should try to do this since it may provide more stable and better averaged nucleosome profiles.

A feature of all active promoter profiles is a strongly singly positioned nucleosome, which is reflected clearly in the average profiles therefore we could not improve on these profiles.

Finally, we understand that the previous version of our manuscript did not explain properly the new observations that it contained so we have made significant revisions to the manuscript. We hope that these revisions addresses the concerns of this reviewer.

Reviewer #3 (Remarks to the Author):

This manuscript describes concepts that have been previously shown including the existence of an unstable H2A.Z nucleosome at the TSS of active genes PMID: 19633671; the generation of high-resolution maps that allow the detection of “unstable” nucleosomes PMID: 27889238, and PMID: 27151365; and the concept of that nucleosome positioning does not change but nucleosome accessibility does during transcription PMID: 27151365 and PMID: 28356342. Based on this, the level of novelty of this study is somehow challenged.

We thank this reviewer for bringing these publications to our attention and indeed, they were referenced. In this revised manuscript, we have modified the introduction and discussion to better consider the implications of our new findings in relation to these previous studies. Regarding the novelty of the work: (1) Concerning the existence of an unstable nucleosome (H2A.Z) at the TSS, this has remained controversial and indeed, the dogma in the literature is still that the TSS is nucleosome depleted. We hope that if this manuscript is published, it will finally resolve this controversy. Further, we not only show the existence of an unstable H2A.Z nucleosome at the TSS but also a stable H2A.Z nucleosome that displays lower levels of gene expression. This suggests that regulating the stability of this nucleosome plays an important role in controlling TSS access. This was not known before. (2) As discussed above, we agree that we did not do a good job in highlighting the novel aspects of this manuscript so we have undertaken a major revision (including a new title). Collectively, the key point is that we have found that H2A.Z has multiple roles in regulating promoter chromatin architecture. This is dependent upon where it is located in the promoter (upstream or downstream of the TSS, at the TSS or at TF binding sites), and whether it exists in a stable or unstable nucleosomal form (see new model in Fig. 8b). Further, what is particularly intriguing is that an active promoter has a stable H2A.Z nucleosome at the +1 position. Conversely, a poorly positioned MNase sensitive H2A.Z nucleosome occupies the +1 position in repressed promoters. In addition, H2A.Z nucleosomes occupy transcription factor binding sites to prevent promiscuous access. Finally, we have uncovered a diversity of different types of active and inactive promoter structures that was previously unknown.

Major concerns:

1- The data presented seems quite preliminary, it provides useful information in several particularities of H2A.Z nucleosomes at promoters as a good starting point but it does not provide solid conclusions on the functionality of the different subclasses of H2A.Z-nucleosomes based on their MNase properties nor in the general nucleosomal mapping architecture at gene promoters. It is not a rounded-up story that deepens into the molecular mechanism of their observations. For example: the cellular models presented could have been better used to study nucleosome changes at promoters of genes that directly change upon TGF β treatment or H2A.Z knock down or during carcinogenesis; but the authors just comment that the promoters have the same clusters for nucleosome positioning and major changes in nucleosome accessibility, but these evidences are not put in the context of gene expression and or transcriptional changes.

We thank this reviewer for this justified criticism. To investigate how promoters of genes that directly change upon TGF β treatment or H2A.Z knock down or during carcinogenesis, in the context of gene expression, we produced alluvial plots. These alluvial plots tracked how inactive or active MNase sensitivity promoter clusters changed upon H2A.Z knockdown (new Fig. 2b), TGF- β treatment (new Fig. 3A) and malignant transformation (new Fig. 3B) to active or inactive promoter clusters, respectively. Gene expression plots are included to show the correlation between each MNase sensitivity cluster and gene expression. GO analyses are now performed to uncover the functional significance of these gene expression changes.

To further uncover the biological significance of the different promoter classes (Fig. 1) and the appearance of sub-nucleosomal DNA fragments at the TSS following H2A.Z knockdown, additional GO

analyses were also performed. Adding to the significance of the different promoter classes identified here, most clusters are associated with different biological functions (new Supplementary Table 2).

2- The authors perform k mean clustering for each cell model and for each specific biological question, this generates clusters that look different despite that they come from the same dataset (Fig 1A versus 1C and 1B versus 1D), this creates lots of confusion for the readers very as it is nearly impossible to be able to see common or integrated patterns among cell models and promoter states, how do the clusters of nucleosome positioning (Fig1A and B) overlap to which clusters of nucleosome accessibility correspond (Fig 1C and D). On top of this, the number of clusters seem randomly chosen, I recommend performing the elbow method to determine the optimal number of clusters.

We apologise for this confusion. It is important to point out that the clusters in Panel A versus C and Panel B versus D, are sorted independently. In other words, the sensitivity plots are not sorted based on the total nucleosome plots. Further, MNase accessibility of specifically positioned nucleosomes is not dependent on where the nucleosome (total or H2A.Z) is positioned. We have now clarified this confusion in the text. To illustrate this, below we have sorted the total nucleosome MCF10A data in the same sort order as the MCF10A MNase-sensitivity data. In clusters with dominant sensitive/resistant nucleosomes (below is a zoomed view of the +2 MNase-resistant nucleosome), we observe that total nucleosome fragments occupy positions across the entire promoter region.

Concerning the elbow method, this approach does not provide a clear optimal number of clusters with the possible exception of two (see below). Two clusters (see Supplementary Fig. 5) produces an active promoter configuration that is already established i.e. a promoter with a +1 and +1 nucleosome. Given this, we took an empirical approach whereby we identified the minimum number of clusters that produced distinct promoter clusters without producing redundant nucleosome promoter arrangements. This empirical approach lead to a choice of seven clusters. To illustrate this point, below also shows a comparison between K=2, K=5, K=7, K=10 and K=15. K=10 or 15 does not produce any major new classes that are not seen in K=7. Therefore, we chose K=7.

It is important to point out (as noted in the discussion) that we are not stating there are precisely seven types of promoter arrangements, but clearly there are a variety of different types (i.e. more than two). It is conceivable that many other types of promoter chromatin arrangements exist that differ in more subtle ways. Perhaps it can be presumed that every single promoter is unique in its own way.

3- For the changes in Fig 1C and D this reviewer asks how quantitative is the MNase mTSS-seq method and what “normalising methods or spike ins” have they used. The differences should be measurable, and statistics need to be applied.

The changes in MNase mTSS-seq nucleosome sensitivity (Fig. 1C and D) are quantitative because it is a ratio of low and high MNase digestion of the same chromatin preparation (thus internally normalised).

Thank you for your suggestion, we have now calculated statistics comparing MNase sensitivity values for exemplar categories of the most highly positioned nucleosomes in Fig. 1c and 1d between MCF10A and each experimental condition. We have reported this result in the figure legend of Figure 1 and we report the details of our analysis in the table below.

Figure 1C Total nucleosome sensitivity				
	Comparison	t-statistic	D of F	p-value
Cluster 6 -1 MRN	10A vs CA1a	-18.317	6733.8	< 2.2e-16
	10A vs TGFb	-13.23	6887.3	< 2.2e-16
	10A vs shZ	-29.224	7025.4	< 2.2e-16
Cluster 4 +1 MRN	10A vs CA1a	-22.594	8410.7	< 2.2e-16
	10A vs TGFb	-14.886	8402.5	< 2.2e-16
	10A vs shZ	-38.05	8552.8	< 2.2e-16

Figure 1D H2A.Z ChIP sensitivity				
	Comparison	t-statistic	D of F	p-value
Cluster 4 -1 MRN	10A vs CA1a	-23.7	5976.6	< 2.2e-16
	10A vs TGFb	-21.975	5933.1	< 2.2e-16
Cluster 2 MSN TSS	10A vs CA1a	43.501	8700.7	< 2.2e-16
	10A vs TGFb	31.583	8205.9	< 2.2e-16

It is also important to highlight that the changes observed in Fig.1c and d can now be explained biologically. This is because the alluvial plots can now account for the major chromatin organisational changes observed in these figures because promoters from one type of MNase sensitivity arrangement move to a different MNase sensitivity cluster following H2A.Z knockdown, TGFβ treatment or malignant transformation i.e. they don't stay the same.

4- The association of these nucleosome maps with gene expression and their dynamics when transcription is altered are very vague. The existence of “poised promoters” is not even mentioned in the whole manuscript and there isn't any clear conclusion of how the “nucleosome architectural map” for cell states, cancer state, transcriptional state or promoter state should look like. In this context, the title of this manuscript does not reflect the conclusions of the study.

The new data from the alluvial plots addresses the association of our nucleosome maps with gene expression and their dynamics when transcription is altered between different cellular states. We agree that the title did not match the conclusions of the study so as noted above, we have changed the title to highlight the new findings of this manuscript.

5-Figure 1C/D and S6. The authors described that H2A.Z-nucleosome constitutes the +1 MNR-nucleosomes however the fact that upon H2A.Z KD there is a clear change on nucleosome properties spanning +1kb/-1kb towards MNR is completely ignored, this observation needs to also be commented by the authors

We thank the reviewer for picking up this interesting observation and have now described this observation (second paragraph, page 9). “Interestingly, in addition to the loss of the +1 MRN, other promoter changes are observed including the loss of MNase sensitivity upstream of the TSS. Therefore, changes throughout this promoter class are observed in the absence of H2A.Z.”

6-The insights on the different types of H2A.Z-nucleosomes according to the promoter transcriptional state could have been much more explored to at least give a better definition of these nucleosomes: are they homotypic, heterotypic; do any of them they have H3.3? Is H2A.Z acetylated in any of these H2A.Z nucleosomes? Is there a difference in H2A.Z isoforms?

We have now clarified in the text that the major isoform expressed is H2A.Z.1 (page 6).

We agree with the reviewer that it would be interesting to further characterize H2A.Z nucleosomes in the context of the current work. However, we are hesitant to undertake these long-term experiments (requiring a considerable amount of time and resources, which may or may not yield readily informative results) for multiple reasons:

1. Concerning heterotypic H2A.Z-H2A nucleosomes, to our knowledge, we were in fact the first to show that they exist on the TSS of active genes in mammalian cell lines (*Nat Struct Mol Biol* **19**, 1076-1083, 2012). However, this analysis was challenging and only performed at a metagene level, not at an individual gene level, because the final amount of H2A.Z ChIP H2A reCHIP DNA recovered is extremely low (because the procedure involves two successive ChIP steps). Therefore, this experiment that this reviewer requests is technically highly demanding i.e. it is not possible to achieve the promoter resolution required to map individual heterotypic nucleosomes on individual promoters even with the 20-fold enrichment using the mTSS-seq method. Instead, we have speculated in the discussion that the TSS MNase sensitive nucleosome is a H2A.Z-H2A heterotypic nucleosome (that contains H3.3, see point 2) based on our and other previous studies.
2. With regards to whether H2A.Z MNase sensitive nucleosomes contain H3.3 or not, these experiments are also problematic. This is because to perform H3.3 ChIP experiments requires a different salt concentration to that used in our study. Specifically, low-ionic-strength buffers are required (*Nature Genetics*, **41**, 941-946, 2009) to detect unstable H3.3 nucleosomes compared to the universal moderate salt conditions that we employed in this manuscript. Therefore, our existing H2A.Z ChIP data would not be comparable to such H3.3 ChIP data. Further, performing H2A.Z ChIP H3.3 re-ChIP compounds this problem (in addition, while we want to base our arguments on scientific reasons, our laboratories have been closed due to Covid-19).
3. Based on our experience, even if we could perform these experiments, we believe that the results would not be clear-cut and could open up an entire catalogue of new questions. Further, these experiments would not alter the major conclusions of this manuscript. Instead, we have addressed point 9 below revealing the histone modifications associated with the different promoter clusters.

7- The plots from Fig 1D are very confusing to interpret and are quite misleading, the MRN give a negative value that it looks like there is a NFR instead of a nucleosome, the authors should overlay (all in positive values) the MSN and MRN data in the same plot.

We agree that the graphs are difficult to interpret. After several trials, we believe colouring the resistant and sensitive nucleosomes blue and yellow, respectively makes the data more understandable.

8- The average gene expression represented as a straight line of a mean across the promoter is unacceptable (Fig. 2b, Fig3a and 3b), please use violin or box plot representations in a separate plot to see the deviation and include p-values.

This was definitely unacceptable (it was our attempt to make the figures easier to interpret). We have now included box plots with p-values.

9- H2A.Z nucleosomes and MNase total heatmaps look almost the same (Fig 1a/b and S4), could authors make a comment on this? Is this expected to occur in all +/- 1kb promoters? I would recommend to have extra controls of other histone variants H3.3 or histone PTMs that are expected to have a discrete localisation in the promoters.

Indeed, this is an important point, that the positioning of bulk nucleosomes and H2A.Z nucleosomes occupy largely the same positions i.e. they are one of the same (which is to be expected). However, there are subtle differences. For example, there are two different H2A.Z promoter classes that occupy the -1 position slightly differently (cluster 3 and 6, Fig. 1b) whereas for total nucleosomes, there is only one -1 promoter class (cluster 6, Fig. 1a). Also, cluster 1 for H2A.Z does not exist in the total nucleosome cluster analysis. Therefore, they are not exactly the same.

Excitingly, as requested, we have examined the different promoter classes identified in our study with publicly available ChIP-Seq data for MCF10A cells obtained from NCBI's Sequence Read Archive (H3K4me1, H3K4me3, H3K79me2, H3K27ac, H2BK20ub, H3K9me3, H3K9ac, H3K36me3, H3K23ac and H4K8ac; new Supplementary Fig. 10). Confirming our new classification of the different types of active and inactive promoter nucleosome arrangements, all active promoter configurations (nucleosome occupancy and MNase sensitivity) display the expected active histone marks.

9. Have the in-house H2A.Z antibody been properly checked for its specificity for ChIP?

Our antibody became the "gold standard" for H2A.Z research in the field (being one of the first mammalian antibodies raised against H2A.Z). We have numerous publications using this antibody (see Methods for a few), it has been given to numerous labs around the globe, and is sold by Millipore (plus periodically, we compare our antibody with other commercial antibodies and have not found an antibody better than ours).

10- Overall, the manuscript and their figures have not been carefully written and made, respectively. The manuscript is very hard to follow, there are many missing labels or misleading labels in the figures (this gets quite bad in the supplementary figures), there is missing information in the material and methods (western blots for KD validation?). In general, the figures are not self-explanatory, the figure legends lack of explanation and clarification of the abbreviations and some terms are used differently when they are the same thing, in example total MNase-seq and mTSS-seq, there are missing references, etc.

We thank the reviewer for his/her thoroughness, and have made the appropriate corrections.

11- Downsampling analyses for comparison between H2A.Z ChIP-seq and H2A.Z mTSS-seq. The authors show a clear improvement of 19X fold when they compare their mTSS-seq and standard H2A.Z ChIP-seq when the latter is down sampled to 2.5 PE million reads. I wonder how the mTSS-seq data performs when it is compared to a H2A.Z-ChIP with standard 45 PE million reads.

Downsampling was necessary to provide a normalisation step in order for the comparison between H2A.Z mTSS-seq (reads only from the promoter) versus standard genome-wide H2A.Z ChIP-seq (reads genome-wide) to be performed. We now have carried out a more thorough bioinformatic analysis (Supplementary Fig. 1) noting that our standard H2A.Z ChIP was sequenced to a depth of 277 PE millions reads. Even with such a high-sequencing depth, the quality of the H2A.Z mTSS profiles are far superior than the standard ChIP profiles (Supplementary Fig. 1a-c).

Basically, we re-ran our subsampling across a range of conditions in order to better estimate the performance of the capture approach. In this instance we chose to compare the TSS region coverage of the TSS-Seq libraries with the same libraries sequenced without TSS capture for three conditions: subsampling 5M and 10M reads, and including the fully sequenced libraries (genome-wide, non-captured libraries sequenced to approximately 277M PE reads).

We simulate the low-coverage experiments by subsampling (using sambamba view -s) each included library to 5 million and 10 million PE reads and also included the fully sequenced libraries. The resulting BAM files were further processed using deeptools (3.2.1) to first calculate the coverage in RPKM across known promoters (1000bp +/- of TSS) and then visualised the coverage using heatmaps (Supplementary Figure 1 a,b,c). In order to illustrate the enrichment effect, we calculated the mean observed coverage in each of the libraries. The results are presented in Supplementary Fig. 1d.

At 5M reads there is an 18.4-fold mean enrichment, which remains near constant at 10M reads (18.3-fold) indicating a linear increase in coverage in both library types. In the fully sequenced libraries, we observed a mean coverage of 14.8 in the capture library versus 1.27 in the conventional library (a 11.65-fold enrichment). However, the capture libraries for H2A.Z ChIP were sequenced to approximately 27M and 30M PE reads versus subsampled 45M PE reads for the conventional H2A.Z ChIP library, indicating a significantly improved TSS coverage in the capture libraries and thus demonstrating the advantage of using this technology for studying specific regions of the genome at very high resolution. Indeed, only at the resolution achieved with the use of the capture technology were we able to distinguish the patterns of nucleosome positioning that underly the different promoter architectures described in this manuscript. These patterns were not discernible in the genome-wide libraries even when using the data from the fully sequenced libraries which yielded 277M reads, and thus are virtually at saturation (data not shown).

Minor comments:

1-The cell model systems need to be better validated; the authors should show the phenotypic and transcriptional effects of TGFB treatment and H2A.Z KD to demonstrate that the cells undergo EMT.

The cell model systems were described by us previously (*Cell Rep* **21**, 943-952 (2017)) where we demonstrated that the knockdown of H2A.Z mimicked TGF β treatment. We have now provided a clearer description of this work in the Introduction.

2- In the sentence “Unexpectedly, a distinct H2A.Z MSN occupies the TSS” the word “unexpectedly” should be removed as this type of nucleosome has been already described PMID: 19633671.

Unexpectedly has been removed.

3- Likewise, the word “unexpectedly” should be removed in the sentence “Unexpectedly, for both total (Fig. 1A) and H2A.Z-containing (Fig. 1B) nucleosomes, we observe different clusters that display a single, high occupancy nucleosomes located at different positions within 500 bp of the TSS.”; a similar concept has been described previously PMID: 27889238.

Unexpectedly has been removed.

4- The word “coverage” is wrongly used when it should be referred as “depth”.

We have changed the word coverage to depth where appropriate (Summary, Discussion and several locations in the Results).

5- In Figure 3, please add expression fold change of the promoters that show increase or decrease subnucleosomal fragments.

We have now included expression box plots to show gene expression (Fig. 3 now has become Fig. 5) showing that the red cluster 1, for all conditions, have the greatest abundance of small DNA fragments and the highest level of expression. Further, as requested, we have included additional data for Fig. 3b. For example, that cluster 1 (red) which has an increase of subnucleosomal fragments in the H2A.Z knockdown condition contains 21.4% of the top significant differentially expressed genes whereas cluster 6 (pink), with a decrease of small DNA fragments, contains only 9.3% differentially expressed genes (page 14).

To further explore the correlation between changes gene in expression and the increase in in SFs following H2A.Z knockdown, we identified common promoters among: 1) the top quartile of promoters with the most MNase sensitive nucleosome at the TSS; 2) the top quartile of promoters that yield the most SFs at the TSS following H2A.Z knockdown and 3) the top 1% of genes whose expression increases following H2A.Z knockdown (see page 14).

Below we plot the absolute difference in gene expression between the MCF10A and the H2A.Z knockdown cells. This shows that the largest difference occur in cluster 1, corresponding to the increase of subnucleosomal fragments in the H2A.Z knockdown cells.

Reviewers' comments:

Reviewer #1 (Remarks to the Author):

The authors have satisfactorily replied to my critical notes and comments.

Reviewer #2 (Remarks to the Author):

The authors have clarified some issues in the original version. However, I still have some questions on the manuscript.

1. The authors identified seven clusters of promoters based on the detected nucleosome patterns. Since the clustering of promoters is critically dependent on the quantitative capturing of nucleosomal fragments by the selected probes located in the promoter regions, the authors need to perform a thorough characterization of the probes regarding their efficiency in capturing the small DNA fragments. The nucleosome signals should be corrected by the efficiency of capture before the clustering analysis. According to the method section, it appears the authors did not do this.

2. An important conclusion of the manuscript is that H2A.Z blocks binding of TFs including FOS and STAT proteins by restricting the target site accessibility. The authors need to provide ATAC-seq data to support this claim.

3. The authors claim that blocks binding of TFs and thus, knockdown of H2A.Z increased small fragments (SFs). To support this, the authors show that 34% of SFs overlap with H2A.Z nucleosomes. However, it also means that 66% of SFs do not overlap with H2A.Z nucleosomes, suggesting the generation of most SFs are not dependent on H2A.Z. The authors need to provide more supporting evidence. Is the SF level positively correlated with H2A.Z level?

Reviewer #3 (Remarks to the Author):

I would like to thank the authors for the throughout revision performed and for the big effort to try to address all the points raised by all reviewers.

I really like the fact that the authors have now properly connected gene expression with the different subtypes of promoters, and their analysis on the changes in promoters associated with changes in gene expression, this has substantially strengthened the manuscript. Given that they have now provided more data that has somehow modified or complemented their conclusions this reviewer still has outstanding questions.

1-Figure 1A: According to the expression analysis shown in Supp Figure 9a, cluster 7 is the cluster with the lowest gene expression, therefore, it is classified as a cluster of inactive promoters and this is stated in the manuscript: "promoter while cluster 7 is overall nucleosome depleted" or "Conversely, low nucleosome occupancy and poor positioning (clusters 7 and 2 for total nucleosome and H2A.Z, respectively) are correlated with low levels of gene expression". The heatmap clearly shows an almost zero signal for MNase-seq, however, this does seem to go against any dogma in epigenetic regulation: nucleosomes act as barriers of transcription, thus overall a promoter that is nucleosome depleted should have high gene expression. On the contrary, the presence of HTPMs at the promoters of cluster 7 is observed, thus suggesting the presence of nucleosomes (Supp Figure 10). The same phenomenon is observed in Figure 1c with cluster 5. Could please the authors explain this in more detail? Is there any technical or analytical reason for this contradictory finding?

If it turns out that the data shows indeed no nucleosome presence, then the authors would need to further prove that this unexpected result is in fact real and not an artifact of the m-TTS-seq technique. I would suggest using alternative methods to footprint nucleosomes at high resolution,

for example, targeted-NOMe-seq.

2- I really like the back to back heatmap comparison between total and sensitivity heatmaps done in the rebuttal for my comment number #2. I would suggest bringing it up to one of the manuscript figures (supplementary) as it would really help the readers to understand nucleosome positioning and nucleosome stability in parallel. It would really make the currently missing link between Figure 1-b and Figure c-d

3- Given the almost identical profiles between H2A.Z and all nucleosomes (Suppl Figure 4), I would also add as a conclusion of the second section of results that all nucleosomes at gene promoters are H2A.Z-nucleosomes.

4- I find that a number of statements throughout the manuscript do not fully correspond with the data shown:

*"Further, it appears that malignant transformation mirrors these H2A.Z-dependent changes in MNase sensitivity suggesting that the knockdown of H2A.Z expression may mimic malignancy (Fig. 1c)": eyeballing the profiles it looks to me that the nucleosomal profile looks more similar to the TGFB condition rather than H2A.Z knockdown, unless a quantitative comparison within each cluster between each condition is performed, this statement is to me inaccurate.

*"Notably, H2A.Z-MSNs are associated with the highest level of gene expression, while the presence of an H2A.Z-MRN at the TSS is correlated with the lowest level of gene expression." If I have understood it correctly, this analysis is focussed only in cluster 2 from figure 1d, so it this statement should be put in the context of cluster 2 from figure 1d, as the levels of expression of clusters 1, 3, 4, 6 are way higher than cluster 2, this needs to be clarified.

*"2) Micrococcal resistance across the promoter but in particular a MRN at the +1 position (total and H2A.Z, colored blue in the average plots) and significantly": according to Figure 2a the +1 MRN is only present in total nucleosomes but not in the H2A.Z one, it reads as the +1 is an H2A.Z nucleosome.

*"Interestingly, the high expressing MSN-TSS cluster displays strong nucleosome positioning (H2A.Z and total) downstream of the TSS" I can see that also happening upstream the TSS.

*"Unexpectedly, upon H2A.Z knockdown, there is a dramatic increase in small sized DNA fragments (SFs) at the TSS." I can only see that happening at the TSS of cluster 1 but not in clusters 1 or 3, please modify this statement to only refer to cluster 1

*"This suggests the H2A.Z-containing nucleosome is further destabilized upon the loss of H2A.Z, resulting in an increase in MNase accessibility." This only occurs at the H2A.Z-MSN nucleosome but not at the H2A.Z-MRN ones, please clarify.

*"In each different experimental condition, cluster 1 contains promoters that have an increase of SFs at the TSS. In TGF- β induced cells, this cluster contains 22% of the top significantly differentially expressed genes as compared to MCF10A... Cluster 6, which represents promoters that contain more SFs in the MCF10A cells, but less SFs than in each experimental condition, contains only 9-12% of the top significantly differentially expressed genes. We conclude that the appearance of SFs at the TSS is an unexpected feature of an active promoter." Do the authors mean that in cluster 1 22% of the top up-regulated genes and cluster 6, 9-12% of the top downregulated genes? Differentially expressed can indicate up or down-regulation so they need to clarify the direction of each cluster as shown in the box plots.

5. Where does H2A.Z-MRN (green cluster 2) from Figure 4 come from on the clustering done in figure 1d? I cannot seem to find any H2AZ-MRN at the TSS.

6. The explanation for the subsequent analysis of H2A.Z at the TSS done in Figure 4 is a bit confusing: "we sorted the H2AZ MNase sensitivity data at +/- 50bp surrounding the TSS to resolve sensitivity patterns more precisely over the TSS." I don't understand, 50bp is smaller than one nucleosome and clearly it is not what it is represented in Figure 4a (-1kb+1kb), in the figure legends it says 100bp. Does figure 4a have all promoters? Why does it look so different to figure 1d?

7. Figure 8 final model: the H2A.Z-MRN nucleosomes at different locations should show the same levels of gene activation as shown in figure 3, with the exception of the H2AZ-MRN at the TSS. I would also add the nucleosomal profile of inactive promoters.

The model including TFs is confusing, according to figure 4, SFs occur at H2A.Z-MSN upon H2A.Z knockdown so it should be an H2A.Z-MSN what is blocking the TF binding.

An additional dynamic model of gene activation upon H2A.Z removal and SFs formation would be nice.

Minor comments:

1. New Figure 2 b and c and Figure 3a-d: please make sure the lines match with the cluster location that they are coming from, some of them are not aligned, for example, cluster 4 or cluster 1 from figure 2c

2. I am wondering if the mixture of MRN nucleosomes (-2, -1, +1, +2) observed when genes become active correspond to different stages of transcription where the well-positioned nucleosome is the leading edge of transcription and the phenomenon observed depends on the gene expression asynchronous of the cells.

3. Fig. 4d: It looks that there is a missing track for the MCF10A+shH2A.Z SFs or the MNase-seq track? I am not sure as the label "MCF10A+shH2A.Z" is confusing.

4. In the discussion, I would also add posttranslational modifications of H2A.Z as an additional way to explain the different levels of H2A.Z nucleosome stability, for example, H2A.Z acetylation could contribute to H2A.Z destabilization.

5. Supp Figure 10 (HPTMs): The correlation of active HPTMs with active promoters is nicely shown, however, their conclusions over inactive promoters are a bit vague: clusters 7, 4 in a), cluster 2 for b); clusters 5 and 2 for c) and clusters 5 and 7 for d): "Conversely, these active modifications are depleted from the different types of inactive promoter chromatin structures that lack positioned and MNase resistant nucleosomes (Supplementary Fig. 10)." I can appreciate in this figure an intermediate enrichment in a 2/3 of the "inactive promoters" of active marks (k4me3, k27ac, k9ac) and also inactive marks k27me3, suggesting a bivalent/poised state, this should be also commented.

6. The justification of 18 fold enrichment is hidden in the supplementary information, so I would recommend extracting some of the info from the suppl figure legend 1 to the main text (especially for Supp Fig 1d) and please refer to subsections of supplementary figure 1 when you explain each part in the main text.

I also recommend to slightly modify suppl figure 1d including the fold change of H2A.Z enrichment at TSSs.

The use of median and mean in the table from supp figure 1d looks very confusing. I am guessing that the mean is calculated around the TSS only and the median is the "middle" value of the overall reads regardless of where they are located?

Reviewers comments

Reviewer 2.

This reviewer has raised new questions and proposed new experiments.

The authors identified seven clusters of promoters based on the detected nucleosome patterns. Since the clustering of promoters is critically dependent on the quantitative capturing of nucleosomal fragments by the selected probes located in the promoter regions, the authors need to perform a thorough characterization of the probes regarding their efficiency in capturing the small DNA fragments. The nucleosome signals should be corrected by the efficiency of capture before the clustering analysis. According to the method section, it appears the authors did not do this.

The sequence capture approach used in this manuscript has been validated and accepted in peer-reviewed publications (e.g. PMID: 26735342, PMID: 26771136, PMID: 32404364) from our group. A thorough characterisation of the selected probes was performed by the authors in a previous publication that used the mTSS-seq approach (PMID 26771136, p6461 and Figure 1B). Data showing the efficiency of the capture is indeed presented in Supplementary Figure 1F of the current manuscript. Moreover, all comparisons are made among similarly derived cell lines and if there were any bias for any particular sequence, this would be present across all cell lines equally and therefore would not alter any conclusion.

1. *An important conclusion of the manuscript is that H2A.Z blocks binding of TFs including FOS and STAT proteins by restricting the target site accessibility. The authors need to provide ATAC-seq data to support this claim.*

We strongly argue that ATAC-Seq will not provide any additional and new information that MNase-Seq does not already provide. Indeed, it is well established that MNase “is a superior method for probing genome-wide nucleosome distributions and also provides an accurate way for assessing transcription factor occupancy” (<https://doi.org/10.1186/1756-8935-7-33>, and also see Ref 28). Further, as this manuscript demonstrates, MNase-Seq provides functional data, which ATAC-seq does not, which is the generation of small sized DNA fragments at transcription factor binding sites. Taken together (and with the extra cost and time involve), we believe that performing ATAC-seq is not justified.

Furthermore, nowhere in the manuscript do we actually state that we “conclude that H2A.Z blocks transcription factor binding”. Our manuscript unequivocally demonstrates that there is an increase in promoter accessibility following H2A.Z knockdown. In the results we “suggest that H2A.Z nucleosomes are located at TFBS to prevent promiscuous transcription factor access”. Likewise, in the discussion, we state “This suggests that H2A.Z has an important role in preventing promiscuous DNA access”. Therefore, we believe that we have not over interpreted the data

2. *The authors claim that blocks binding of TFs and thus, knockdown of H2A.Z increased small fragments (SFs). To support this, the authors show that 34% of SFs overlap with H2A.Z nucleosomes. However, it also means that 66% of*

SFs do not overlap with H2A.Z nucleosomes, suggesting the generation of most SFs are not dependent on H2A.Z. The authors need to provide more supporting evidence. Is the SF level positively correlated with H2A.Z level?

As our high stringency bioinformatic analyses revealed that the reason that only 34% of small DNA fragments overlap with H2A.Z nucleosomes is because it is only those H2A.Z nucleosomes that sit on transcription factor binding sites produce small DNA fragments i.e. not all H2A.Z nucleosomes sit on transcription factor binding sites. As described in the results, it is only this population of H2A.Z nucleosomes that produce small DNA fragments.

Reviewer 3

Major Points

We thank this reviewer for acknowledging our efforts to address all the reviewer's comments.

1. *Figure 1A: According to the expression analysis shown in Supp Figure 9a, cluster 7 is the cluster with the lowest gene expression, therefore, it is classified as a cluster of inactive promoters and this is stated in the manuscript: "promoter while cluster 7 is overall nucleosome depleted" or "Conversely, low nucleosome occupancy and poor positioning (clusters 7 and 2 for total nucleosome and H2A.Z, respectively) are correlated with low levels of gene expression". The heatmap clearly shows an almost zero signal for MNase-seq, however, this does seem to go against any dogma in epigenetic regulation: nucleosomes act as barriers of transcription, thus overall a promoter that is nucleosome depleted should have high gene expression. On the contrary, the presence of HTPMs at the promoters of cluster 7 is observed, thus suggesting the presence of nucleosomes (Supp Figure 10). The same phenomenon is observed in Figure 1c with cluster 5. Could please the authors explain this in more detail? Is there any technical or analytical reason for this contradictory finding?*

If it turns out that the data shows indeed no nucleosome presence, then the authors would need to further prove that this unexpected result is in fact real and not an artifact of the m-TTS-seq technique. I would suggest using alternative methods to footprint nucleosomes at high resolution, for example, targeted-NOMe-seq.

We understand, and apologise for this confusion with our description and terminology, which clearly needs further clarification. The information that the heat maps provide is whether there is a common or different nucleosome positioning pattern for different types of promoters based on nucleosome abundance at the -2, -1, +1 and +2 positions. However, this type of analysis does not provide information on the overall nucleosome abundance (density) at promoters and therefore, the low signal of MNase-Seq observed in cluster 7 indicates that no nucleosomes across the promoter occupy the same positions in this cluster. Indeed, this is exemplified in Supplementary Figure 5. When a clustering analysis is performed with K=2, about 80% of promoters display a close to zero MNase-Seq signal. However, when K=7 (Figure 1), it becomes clear that this 80% population of promoters is comprised of many different types of positioned nucleosome arrangements. This illustrates that it is only nucleosome positioning information that these

heat maps provide and not overall nucleosome abundance. Clarification of the information that the heat maps provide is now included on page 7, “The information that the heat maps provide is whether there is a common or different nucleosome positioning pattern for different types of promoter classes based on the nucleosome abundance at the -2, -1, +1 and +2 positions”.

Therefore, it was not our intention to imply that the promoter arrangement in cluster 7 is in general or is overall nucleosome deficient or depleted. What we wanted to convey is that there is an overall depletion of high occupancy positioned nucleosomes. We have simplified the text and stated the following: “the major feature of cluster 7 is the absence of any clear nucleosomal organisation pattern that displays strongly positioned nucleosomes”, page 7; Conversely, this “poor nucleosome positioning is correlated with low levels of gene expression”, page 11. In other words, the lack of strongly positioned nucleosomes in the promoter is associated with gene repression. Indeed, this does not go against the dogma because, as this manuscript and other studies have shown, strongly positioned nucleosomes are a feature of active promoters.

Second, as discussed above, the almost zero signal of MNase-Seq is not because of the absence of nucleosomes but because there is an absence of nucleosomes that are positioned with any uniformity in this cluster. This random arrangement of these nucleosomal DNA fragments can indeed be observed in cluster 7 at a higher zoom magnification. Third, therefore, given that nucleosomes are present in this cluster, and the top quartile of promoters are moderately expressed (Fig. S10), this is consistent with the presence of HTPMs. Cluster 5 in Figure 5C is based on Mnase sensitivity and the lack of signal is because this cluster is neither resistant nor sensitive and therefore is not related to nucleosome occupancy. Finally, our observations are concordant with other studies of nucleosome distribution and gene expression, in which low occupancy disordered nucleosome architecture is associated with lower gene expression, see PMID: 28356342 and Figure 1C.

2. *I really like the back to back heatmap comparison between total and sensitivity heatmaps done in the rebuttal for my comment number #2. I would suggest bringing it up to one of the manuscript figures (supplementary) as it would really help the readers to understand nucleosome positioning and nucleosome stability in parallel. It would really make the currently missing link between Figure 1-b and Figure c-d*

We have now included the heatmap comparison between total and sensitivity heat maps (new Supplementary 6, description of this figure is in page 9).

3. *Given the almost identical profiles between H2A.Z and all nucleosomes (Suppl Figure 4), I would also add as a conclusion of the second section of results that all nucleosomes at gene promoters are H2A.Z-nucleosomes.*

We cannot make the conclusion that all promoter nucleosomes are H2A.Z-nucleosomes. What the data indicates is that canonical nucleosomes and H2A.Z-containing nucleosomes occupy the same

positions. Moreover, it is unlikely that any promoter nucleosome is a “100%” H2A.Z nucleosome because this is dependent upon the exchange dynamics of the SWR1 H2A.Z deposition complex.

4. 1) *“Further, it appears that malignant transformation mirrors these H2A.Z-dependent changes in MNase sensitivity suggesting that the knockdown of H2A.Z expression may mimic malignancy (Fig. 1c): eyeballing the profiles it looks to me that the nucleosomal profile looks more similar to the TGFB condition rather than H2A.Z knockdown, unless a quantitative comparison within each cluster between each condition is performed, this statement is to me inaccurate.*

The statement that the knockdown of H2A.Z expression may mimic malignancy has been removed.

“Notably, H2A.Z-MSNs are associated with the highest level of gene expression, while the presence of an H2A.Z-MRN at the TSS is correlated with the lowest level of gene expression.” If I have understood it correctly, this analysis is focussed only in cluster 2 from figure 1d, so it this statement should be put in the context of cluster 2 from figure 1d, as the levels of expression of clusters 1, 3, 4, 6 are way higher than cluster 2, this needs to be clarified.

“Notably, H2A.Z-MSNs are associated with the highest level of gene expression” was not focused on cluster 2 in Figure 1d. All ~22,000 promoters were sorted based on the H2AZ MNase sensitivity data at the TSS, which revealed a new promoter class containing a H2A.Z-MRN nucleosome at the TSS. We have qualified (on page 13) that H2A.Z-MSN nucleosomes have the highest level of expression in relation to the other two TSS promoter classes in Fig. 4a (those that lack a H2A.Z nucleosome at the TSS and those that have a MRN at the TSS).

“(2) Micrococcal resistance across the promoter but in particular a MRN at the +1 position (total and H2A.Z, coloured blue in the average plots) and significantly”: according to Figure 2a the +1 MRN is only present in total nucleosomes but not in the H2A.Z one, it reads as the +1 is an H2A.Z nucleosome.

The +1 H2A.Z nucleosome is resistant i.e. the MNase sensitivity is -0.6 but not as resistant as the total nucleosome (-1.2). We now realise that the problem was that we did omit to colour the +1 H2A.Z nucleosome blue in Figure 2a.

“Interestingly, the high expressing MSN-TSS cluster displays strong nucleosome positioning (H2A.Z and total) downstream of the TSS” I can see that also happening upstream the TSS.

We have added [and upstream] of the TSS (on page 13) to this sentence.

“Unexpectedly, upon H2A.Z knockdown, there is a dramatic increase in small sized DNA fragments (SFs) at the TSS.” I can only see that happening at the TSS of cluster 1 but not in clusters 1 or 3, please modify this statement to only refer to cluster 1

We have changed “Unexpectedly, upon H2A.Z knockdown, there is a dramatic increase in small sized DNA fragments (SFs) at the TSS” to “Unexpectedly, upon H2A.Z knockdown, there is a dramatic increase in small sized DNA fragments (SFs) at the TSS at cluster 1, and a modest increase at cluster 2” (Page 13). For cluster 2 (Fig. 4c), it is important to directly compare the MNase Seq data with the shH2A.Z data and then it is clear that there is a small increase in small DNA fragments for this cluster.

**“This suggests the H2A.Z-containing nucleosome is further destabilized upon the loss of H2A.Z, resulting in an increase in MNase accessibility.” This only occurs at the H2A.Z-MSN nucleosome but not at the H2A.Z-MRN ones, please clarify.*

We have changed “This suggests the H2A.Z-containing nucleosome is further destabilized upon the loss of H2A.Z, resulting in an increase in MNase accessibility” to “This suggests the H2A.Z-containing nucleosome, in particular the H2A.Z-MSN, is further destabilized upon the loss of H2A.Z, resulting in an increase in MNase accessibility” (page 13).

**“In each different experimental condition, cluster 1 contains promoters that have an increase of SFs at the TSS. In TGF- β induced cells, this cluster contains 22% of the top significantly differentially expressed genes as compared to MCF10A... Cluster 6, which represents promoters that contain more SFs in the MCF10A cells, but less SFs than in each experimental condition, contains only 9-12% of the top significantly differentially expressed genes. We conclude that the appearance of SFs at the TSS is an unexpected feature of an active promoter.” Do the authors mean that in cluster 1 22% of the top up-regulated genes and cluster 6, 9-12% of the top downregulated genes? Differentially expressed can indicate up or down-regulation so they need to clarify the direction of each cluster as shown in the box plots.*

We have corrected this error by removing “We conclude that the appearance of SFs at the TSS is an unexpected feature of an active promoter” because this differential analysis was for genes whose expression significantly went up or down as pointed out by this reviewer. This sentence has been added to the preceding paragraph because this observation is based on the unequivocal results of Fig. 5a (we placed this conclusion in the wrong position in the text) (page 14).

5. *Where does H2A.Z-MRN (green cluster 2) from Figure 4 come from on the clustering done in figure 1d? I cannot seem to find any H2AZ-MRN at the TSS.*

The window over which the clustering analysis is focused drives the clustering analysis. In Figure 1, we were interested in general principles of nucleosome architecture, and focused on the four nucleosomes surrounding the transcription start site. Our clustering window was set to +/- 500 bp surrounding the TSS. Even though we used only the information for +/- 500 bp surrounding the TSS, we show the entire window studied, the 2000 bp surrounding the TSS. In Figure 4, we were interested in the occupancy directly over the TSS; therefore, we focused

our clustering analysis on the TSS +/- 50 bp, a 100 bp window. Even though we focused our analysis on the 100 bp flanking the TSS, the fragments that we identified at this area are nucleosomal sized DNA fragments size (as shown in the total MNase digestion fragment size distribution analysis in Figure 4C where a 145 bp DNA fragment is present on the TSS).

6. *The explanation for the subsequent analysis of H2A.Z at the TSS done in Figure 4 is a bit confusing: “we sorted the H2AZ MNase sensitivity data at +/- 50bp surrounding the TSS to resolve sensitivity patterns more precisely over the TSS.” I don’t understand, 50bp is smaller than one nucleosome and clearly it is not what it is represented in Figure 4a (-1kb+1kb), in the figure legends it says 100bp. Does figure 4a have all promoters? Why does it look so different to figure 1d?*

Following on from point 5, the standard sorting algorithm applied does not discriminate on size i.e. captures all sized DNA fragments including 145bp nucleosomal DNA fragments (as shown by the phasing of H2A.Z and total nucleosomes upstream and downstream of the TSS in Figure 4a) for the entire promoter region (+/- 1kb) with the sort centred at the TSS. By focusing the sort at the TSS, this revealed the presence of H2A.Z micrococcal resistant and sensitive nucleosomes at the TSS (and why the former was not seen in Fig.1d).

7. *Figure 8 final model: the H2A.Z-MRN nucleosomes at different locations should show the same levels of gene activation as shown in figure 3, with the exception of the H2AZ-MRN at the TSS. I would also add the nucleosomal profile of inactive promoters. The model including TFs is confusing, according to figure 4, SFs occur at H2A.Z-MSN upon H2A.Z knockdown so it should be an H2A.Z-MSN what is blocking the TF binding. An additional dynamic model of gene activation upon H2A.Z removal and SFs formation would be nice.*

We have modified the Figure 8 model as requested, we have :1) correlated each promoter configuration according to its level of expression, 2) added the nucleosomal profile for an inactive promoter and 3) show now that a MSN blocks the binding of TFs. We understand that showing a dynamic model would be nice. However, given that the major finding of this manuscript is that H2A.Z displays multiple positions and roles in regulating promoter chromatin organisation, we found that it was not possible to capture all the key findings in a simple model. Therefore, we prefer to remain close to the data and only summarise the major observations as depicted in Figure 8a.

Minor points

1. *New Figure 2 b and c and Figure 3a-d: please make sure the lines match with the cluster location that they are coming from, some of them are not aligned, for example, cluster 4 or cluster 1 from figure 2c*

The alluvial plots have been fixed.

2. *I am wondering if the mixture of MRN nucleosomes (-2, -1, +1, +2) observed when genes become active correspond to different stages of transcription where the well-positioned nucleosome is the leading edge of transcription and the phenomenon observed depends on the gene expression asynchronous of the cells.*

The majority of the cells in the population are in G1 (~62%) and therefore we do not believe the mixture of MRN nucleosomes can be attributed to an asynchronous population of cells.

3. *Fig. 4d: It looks that there is a missing track for the MCF10A+shH2A.Z SFs or the MNase-seq track? I am not sure as the label "MCF10A+shH2A.Z" is confusing.*

In Fig. 4d, there is no missing track. This track shows that there are absolutely no small DNA fragments in wt. MCF-10A cells but a major increase following the knockdown of H2A.Z expression. Additionally, we have changed the label for the H2A.Z knockdown track to simply shH2A.Z.

4. *In the discussion, I would also add posttranslational modifications of H2A.Z as an additional way to explain the different levels of H2A.Z nucleosome stability, for example, H2A.Z acetylation could contribute to H2A.Z destabilization.*

It was an oversight not to include H2A.Z acetylation (which was in the original manuscript). This has now been added in the discussion.

5. *Supp Figure 10 (HPTMs): The correlation of active HPTMs with active promoters is nicely shown, however, their conclusions over inactive promoters are a bit vague: clusters 7, 4 in a), cluster 2 for b); clusters 5 and 2 for c) and clusters 5 and 7 for d): "Conversely, these active modifications are depleted from the different types of inactive promoter chromatin structures that lack positioned and MNase resistant nucleosomes (Supplementary Fig. 10)." I can appreciate in this figure an intermediate enrichment in a 2/3 of the "inactive promoters" of active marks (k4me3, k27ac, k9ac) and also inactive marks k27me3, suggesting a bivalent/poised state, this should be also commented.*

We do not believe the intermediate enrichment of active histone marks is due to a bivalent state (and moreover, given that each repressed cluster contains thousands of genes, it is not possible to make such a conclusion). For the strongly repressed promoter classes (clusters 2 and 3, and cluster 5, now Supp Fig. 11 c and d, respectively), there is a complete absence of active marks. For the moderately repressed promoters (clusters 5 and 7, Fig. 11 c and d, respectively), there is an intermediate level of active marks is observed. This is because, as the expression box plots show, not all promoters are completely repressed in these promoter clusters i.e. Fig. 2d and 3e show that while the median expression of these intermediate repressed clusters is very low, the top quartile of genes are expressed. This has been clarified on page 13.

6. *The justification of 18-fold enrichment is hidden in the supplementary information, so I would recommend extracting some of the info from the suppl figure legend 1 to the main text (especially for Supp Fig 1d) and please refer to subsections of supplementary figure 1 when you explain each part in the main text.*

I also recommend to slightly modify suppl figure 1d including the fold change of H2A.Z enrichment at TSSs.

The use of median and mean in the table from supp figure 1d looks very confusing. I am guessing that the mean is calculated around the TSS only and the median is the “middle” value of the overall reads regardless of where they are located?

Supplementary Fig. 1d now includes the fold change of enrichment. Median and mean values are calculated for coverage in 10bp windows across all defined promoter regions (this description has been added to the figure legend of Supp. Fig. 1d). Therefore, the values shown in Supplemental Figure 1d are observed across all promoter regions as defined by the TSS-capture array.

Reviewers' comments:

Reviewer #2 (Remarks to the Author):

In my previous comments, I suggested the authors to perform ATAC-seq experiments to further support one major conclusion in their manuscript, which referred to the sentence in the abstract "Further, the loss of H2A.Z leads to a dramatic increase in the accessibility of transcription factor binding sites." This is an important conclusion in the manuscript and it deserves further validation using a different strategy, in particular if it were to be published in a reputable journal such as Nature Communications. The authors rejected the suggestions by arguing that "MNase is a superior method for probing genome-wide nucleosome distributions and also provides an accurate way for assessing transcription factor occupancy". However, the conclusion is about "a dramatic increase in the accessibility of transcription factor binding sites", which needs to be validated with a different method. ATAC-seq is a well-established method for measuring accessibility of chromatin and thus I suggested it. But the authors may also use other strategies, such as DNase-seq. I feel strongly that important conclusions need to be validated by different methods.

Reviewer #3 (Remarks to the Author):

I would like to thank the authors for their throughout explanation and justification of all of my concerns and for taking on most of my suggestions. The manuscript now reads well and the interpretation of the data is accurate and properly explained.

Reviewer #4 (Remarks to the Author):

1. I do not believe that ATAC-seq data are necessary to support the authors' conclusions, although they would be welcome. ATAC-seq data generally correlate with nucleosome-depleted/free regions (NDRs/NFRs) at promoters, which are defined by MNase-seq data. In this study, it's not clear what the prediction is for ATAC-seq data, since an MNase-sensitive nucleosome may or may not also be sensitive to Tn5 transposase (the enzyme used in ATAC-seq). The result would not be definitive, since the authors could explain it either way.

2. A related issue is the question of how widespread are the H2A.Z-containing MNase-sensitive nucleosomes proposed to occupy the TSS? In Fig. 4a, they appear to be almost undetectable in the H2A.Z ChIP-seq heat map: there is virtually no H2A.Z signal over the TSS in cluster 1 and, in the average profiles, cluster 1 does not show a peak in the NDR. This is presumably because the data shown are for high MNase digestion, after which any MNase-sensitive nucleosomes would have been destroyed. If so, the heat map and profile for the low level MNase digestion should also be shown.

The difference between expression levels for the 3 clusters (Fig. 4b) seems marginal, with very similar medians and means.

3. The authors should state clearly whether the average profiles are midpoint plots or occupancy/coverage plots. I think they are midpoint plots. A high peak in a midpoint plot indicates strong positioning (coincidence of midpoints) but not necessarily high occupancy - weak positioning results in a series of small midpoint peaks which would be spread along the x-axis, but the occupancy might still be high. If occupancy is really plotted here, a very high peak, as seen in many of the plots, would indicate that the regions flanking that peak are virtually nucleosome-free, which is very unlikely. The authors should show coverage plots if they want to argue for high occupancy.

Minor comments:

4. Fig. 2a: Incorrect labels for the two heat maps at left (top and bottom labels are switched).

5. Fig. 7c: The two lefthand motifs are essentially the same (just opposite strands).

6. Ideally, a picture of a typical gel showing the levels of MNase digestion corresponding to high

and low would be included as a supplementary figure. The extent of digestion is a critical variable.

REVIEWER COMMENTS

Reviewer #2 (Remarks to the Author):

In my previous comments, I suggested the authors to perform ATAC-seq experiments to further support one major conclusion in their manuscript, which referred to the sentence in the abstract "Further, the loss of H2A.Z leads to a dramatic increase in the accessibility of transcription factor binding sites." This is an important conclusion in the manuscript and it deserves further validation using a different strategy, in particular if it were to be published in a reputable journal such as Nature Communications. The authors rejected the suggestions by arguing that "MNase is a superior method for probing genome-wide nucleosome distributions and also provides an accurate way for assessing transcription factor occupancy". However, the conclusion is about "a dramatic increase in the accessibility of transcription factor binding sites", which needs to be validated with a different method. ATAC-seq is a well-established method for measuring accessibility of chromatin and thus I suggested it. But the authors may also use other strategies, such as DNase-seq. I feel strongly that important conclusions need to be validated by different methods.

We respectfully disagree with the reviewer for the following reasons (1) MNase sensitivity and its relationship to genomic accessibility has been established; and, (2) it will be impossible for other experimental assays (e.g. ATAC-seq, or DNaseI) to yield a dispositive result.

- (1) The utility of MNase titrations to identify biochemically distinct nucleosomal footprints that are sensitive or resistant to MNase is a decade old (Weiner et. al, 2010; Henikoff et. al, 2011; Xi et. al, 2011). Soon after these initial observations were made, our lab was the first to apply this approach to a multicellular organism (Vera et. al., 2014). Indeed, *Nature Communications* has been at the forefront of publishing manuscripts that have pioneered this approach, using "accessibility" in the title of a foundational manuscript (Mieczkowski et. al., 2016, in which an experimental comparison to ATAC-seq was not presented). The idea of a link between MNase sensitivity and increased genomic accessibility is approaching settled science as evidenced by several papers in leading journals (Mueller et. al, 2017; Chereji et. al., 2019; Jordan et. al., 2020; Parvathaneni et. al., 2020; Lion et. al, 2020; Zhao et. al, 2020; Gao et. al, 2020).

(2) No single experiment is the *sine qua non* determinant of accessibility. ATAC-seq is a good experiment, but different experiments arrive at similar conclusions (DNaseI, MNase, ATAC) without the need for orthogonal work, which may or may not be measuring the same biochemical property. In fact, a recent *Genome Biology* paper (Zhao et. al., 2020) demonstrates this very point. These authors report that while a majority of accessible genomic loci characterized by of MNase, DNaseI, and ATAC-seq are concordant with one another, there are several MNase accessible loci unique to MNase. The authors hypothesize that the small size of the MNase enzyme relative to DNase I or Tn5 is responsible for the identification of these MNase specific sites. Similarly, a systematic comparison of assay specific features across three chromatin accessibility experiments: NOME-, ATAC-, and DNase I-seq (Nordstrom et. al., 2019, see Figure 4). These results suggest that agreement or disagreement with any of these experiments could be explained by assay specific differences and would not alter the conclusions of the manuscript. Finally, reviewers and editors understand that each approach may arrive at its own set of conclusions independent of experiments using other enzymes. For example, *Nature Communications* very recently published (Murphy et. al., 2020), an interesting paper whose conclusions are based solely on ATAC-seq. This manuscript makes conclusions about nucleosome positioning, but was not buttressed by MNase-derived maps.

Reviewer #3 (Remarks to the Author):

I would like to thank the authors for their throughout explanation and justification of all of my concerns and for taking on most of my suggestions. The manuscript now reads well and the interpretation of the data is accurate and properly explained.

We would like to thank Reviewer #3 whose constructive comments have made the manuscript stronger.

Reviewer #4 (Remarks to the Author):

1. I do not believe that ATAC-seq data are necessary to support the authors' conclusions, although they would be welcome. ATAC-seq data generally correlate with nucleosome-depleted/free regions (NDRs/NFRs) at promoters, which are defined by MNase-seq data. In this study, it's not clear what the prediction is for ATAC-seq data, since an MNase-sensitive nucleosome may or may not also be sensitive to Tn5 transposase (the enzyme used in ATAC-seq). The result would not be definitive, since the authors could explain it either way.

We thank the reviewer for this thoughtful assessment, and are in complete agreement. Please see our response above to Reviewer #2 for further detail.

2. A related issue is the question of how widespread are the H2A.Z-containing MNase-sensitive nucleosomes proposed to occupy the TSS? In Fig. 4a, they appear to be almost undetectable in the H2A.Z ChIP-seq heat map: there is virtually no H2A.Z signal over the TSS in cluster 1 and, in the average profiles, cluster 1 does not show a peak in the NDR. This is presumably because the data shown are for high MNase digestion, after which any MNase-sensitive nucleosomes would have been destroyed. If so, the heat map and profile for the low level MNase digestion should also be shown.

The difference between expression levels for the 3 clusters (Fig. 4b) seems marginal, with very similar medians and means.

We thank the reviewer for making this point, as it highlights a major strength of studies such as ours: that sensitivity maps identify an important role for H2A.Z at the TSS that simply would not be detected in nucleosome occupancy maps (this important point is now highlighted in the discussion). The data in Figure 4A are from the combined heavy and light MNase digestions. Yes, relative to the expected high occupancy +1 and -1 nucleosomes, the TSS localized H2A.Z nucleosomes demonstrate a reduced occupancy. The average plot below the 4A total H2A.Z heatmap, however, shows an increased occupancy of the H2A.Z-containing nucleosome at the TSS (cluster 1 red line, bump at TSS). Additionally, the fragment size plots in Figure 4C for cluster 1 (red) show that the occupancy is nucleosome-sized in the H2A.Z plot (bottom left). This nucleosome sized occupancy gives way to smaller sized protections in the H2A.Z sh knockdown. The sensitivity experiments provide the critical discovery that would never have been revealed by looking solely at occupancy data. Using the sensitivity data, we are able to ascribe a function to this low occupancy nucleosome, and conclude that even though this position is not highly occupied, this position is critical to the maintenance of appropriate expression of these genes.

differences in gene expression among the three different promoter classifications

Regarding the gene expression in 4B, these basal MCF10A categories contain thousands of genes and have been shown to be significantly different from one

another. In other words, what appears to be marginal differences in the expression of thousands of genes does accurately reflect the differences among the three different promoter classifications. The effect of this TSS H2A.Z nucleosome is further strengthened in: (1) Figure 4C where the H2A.Z nucleosome sized occupancy gives way to smaller sized protections in the H2A.Z sh knockdown, and (2) Figure 5B in which this increase in small fragments in the TSS in the shZ cells is associated with the highest gene expression (5B right panel, cluster 1).

3. The authors should state clearly whether the average profiles are midpoint plots or occupancy/ coverage plots. I think they are midpoint plots. A high peak in a midpoint plot indicates strong positioning (coincidence of midpoints) but not necessarily high occupancy - weak positioning results in a series of small midpoint peaks which would be spread along the x-axis, but the occupancy might still be high. If occupancy is really plotted here, a very high peak, as seen in many of the plots, would indicate that the regions flanking that peak are virtually nucleosome-free, which is very unlikely. The authors should show coverage plots if they want to argue for high occupancy.

We thank the reviewer for identifying this oversight. Indeed we have plotted nucleosome midpoints +/- 30 bp (i.e. the 60 central nucleosome bp). Therefore, we are indeed looking at both position and occupancy. We have edited the Methods to accurately reflect this point.

Minor comments:

4. Fig. 2a: Incorrect labels for the two heat maps at left (top and bottom labels are switched).

This has been corrected in the Figure.

5. Fig. 7c: The two lefthand motifs are essentially the same (just opposite strands).

We had noticed the similarity between the two motifs at the time we completed the experiment. We realized that even though the motifs are highly similar, slight differences between the motifs and the identified binding factors warranted inclusion of both as separate entities. Further, it is important to point out that we are reporting

the output of the long-established MEME pipeline. Therefore, others could run our small fragments through this pipeline and get the same results.

6. Ideally, a picture of a typical gel showing the levels of MNase digestion corresponding to high and low would be included as a supplementary figure. The extent of digestion is a critical variable.

This experiment showing the degree of digestion was included in Supplementary Figure 1e.

CITATIONS

Buenrostro, Jason D., Paul G. Giresi, Lisa C. Zaba, Howard Y. Chang, and William J. Greenleaf. 2013. "Transposition of Native Chromatin for Fast and Sensitive Epigenomic Profiling of Open Chromatin, DNA-Binding Proteins and Nucleosome Position." *Nature Methods* 10 (12): 1213–18.

Chereji, Razvan V., Terri D. Bryson, and Steven Henikoff. 2019. "Quantitative Measurement of Nucleosome Occupancy and DNA Accessibility." *Biophysical Journal*. <https://doi.org/10.1016/j.bpj.2018.11.436>.

Gao, Weiwu, Binbin Lai, Bing Ni, and Keji Zhao. 2020. "Genome-Wide Profiling of Nucleosome Position and Chromatin Accessibility in Single Cells Using scMNase-Seq." *Nature Protocols* 15 (1): 68–85.

Henikoff, Jorja G., Jason A. Belsky, Kristina Krassovsky, David M. MacAlpine, and Steven Henikoff. 2011. "Epigenome Characterization at Single Base-Pair Resolution." *Proceedings of the National Academy of Sciences of the United States of America* 108 (45): 18318–23.

Jordan, Katherine W., Fei He, Monica Fernandez de Soto, Alina Akhunova, and Eduard Akhunov. 2020. "Differential Chromatin Accessibility Landscape Reveals Structural and Functional Features of the Allopolyploid Wheat Chromosomes." *Genome Biology* 21 (1): 176.

Lion, Mattia, Breynev Muhire, Yuka Namiki, Michael Y. Tolstorukov, and Marjorie A. Oettinger. 2020. "Alterations in Chromatin at Antigen Receptor Loci Define Lineage Progression during B Lymphopoiesis." *Proceedings of the National Academy of Sciences of the United States of America* 117 (10): 5453–62.

Mueller, Britta, Jakub Mieczkowski, Sharmistha Kundu, Peggy Wang, Ruslan Sadreyev, Michael Y. Tolstorukov, and Robert E. Kingston. 2017. "Widespread Changes in Nucleosome Accessibility without Changes in Nucleosome Occupancy during a Rapid Transcriptional Induction." *Genes & Development* 31 (5): 451–62.

Murphy, Kristin E., Fanju W. Meng, Claire E. Makowski, and Patrick J. Murphy. 2020. "Genome-Wide Chromatin Accessibility Is Restricted by ANP32E." *Nature Communications* 11 (1): 5063.

Nordström, Karl J V, Florian Schmidt, Nina Gasparoni, Abdulrahman Salhab, Gilles Gasparoni, Kathrin Kattler, Fabian Müller, Peter Ebert, Ivan G Costa, Nico Pfeifer, Thomas Lengauer, Marcel H Schulz, Jörn Walter. 2019. "Unique and assay specific features of NOME-, ATAC- and DNase I-seq data." *Nucleic Acids Res.* 47(20):10580-10596.

Parvathaneni, Rajiv K., Edoardo Bertolini, Md Shamimuzzaman, Daniel Vera, Pei-Yau Lung, Brian R. Rice, Jinfeng Zhang, et al. n.d. "The Regulatory Landscape of Early Maize Inflorescence Development." <https://doi.org/10.1101/870378>.

Stalder, J., A. Engel Larsen, and Dolan Jd. n.d. "M. Groudine, M. and Weintraub. H. 1980." Tissue-Specific DNA Cleavages in the Globin Chromatin Domain Introduced by DNase I. *Cell* 20: 451–60.

Vera, Daniel L., Thelma F. Madzima, Jonathan D. Labonne, Mohammad P. Alam, Gregg G. Hoffman, S. B. Girimurugan, Jinfeng Zhang, Karen M. McGinnis, Jonathan H. Dennis, and Hank W. Bass. 2014. "Differential Nuclease Sensitivity Profiling of Chromatin Reveals Biochemical Footprints Coupled to Gene Expression and Functional DNA Elements in Maize." *The Plant Cell*. <https://doi.org/10.1105/tpc.114.130609>.

Weiner, Assaf, Amanda Hughes, Moran Yassour, Oliver J. Rando, and Nir Friedman. 2010. "High-Resolution Nucleosome Mapping Reveals Transcription-Dependent Promoter Packaging." *Genome Research* 20 (1): 90–100.

Xi, Yuanxin, Jianhui Yao, Rui Chen, Wei Li, and Xiangwei He. 2011. "Nucleosome Fragility Reveals Novel Functional States of Chromatin and Poises Genes for Activation." *Genome Research* 21 (5): 718–24.

Zhao, Hainan, Wenli Zhang, Tao Zhang, Yuan Lin, Yaodong Hu, Chao Fang, and Jiming Jiang. 2020. "Genome-Wide MNase Hypersensitivity Assay Unveils Distinct Classes of Open Chromatin Associated with H3K27me3 and DNA Methylation in *Arabidopsis Thaliana*." *Genome Biology* 21 (1): 24.